# Clarin-2 is essential for hearing by maintaining stereocilia integrity and function

Lucy A Dunbar[1], Pranav Patni[2], Carlos Aguilar[1], Philomena Mburu[1], Laura Corns[3], Helena RR Wells[4], Sedigheh Delmaghani[2], Andrew Parker[1], Stuart Johnson[3], Debbie Williams[1], Christopher T Esapa[1], Michelle M Simon[1], Lauren Chessum[1], Sherylanne Newton[1], Joanne Dorning[1], Prashanthini Jeyarajan[1], Susan Morse[1], Andrea Lelli[5], Gemma F Codner[6], Thibault Peineau[7], Suhasini R Gopal[8], Kumar N Alagramam[8], Ronna Hertzano[9], Didier Dulon[7], Sara Wells[6], Frances M Williams[4], Christine Petit[5], Sally J Dawson[10], Steve DM Brown[1], Walter Marcotti[3] (iD), Aziz El-Amraoui[2,*,†] (iD) & Michael R Bowl[1,†,**] (iD)

## Abstract

Hearing relies on mechanically gated ion channels present in the actin-rich stereocilia bundles at the apical surface of cochlear hair cells. Our knowledge of the mechanisms underlying the formation and maintenance of the sound-receptive structure is limited. Utilizing a large-scale forward genetic screen in mice, genome mapping and gene complementation tests, we identified *Clrn2* as a new deafness gene. The *Clrn2clarinet/clarinet* mice (p.Trp4* mutation) exhibit a progressive, early-onset hearing loss, with no overt retinal deficits. Utilizing data from the UK Biobank study, we could show that *CLRN2* is involved in human non-syndromic progressive hearing loss. Our in-depth morphological, molecular and functional investigations establish that while it is not required for initial formation of cochlear sensory hair cell stereocilia bundles, clarin-2 is critical for maintaining normal bundle integrity and functioning. In the differentiating hair bundles, lack of clarin-2 leads to loss of mechano-electrical transduction, followed by selective progressive loss of the transducing stereocilia. Together, our findings demonstrate a key role for clarin-2 in mammalian hearing, providing insights into the interplay between mechano-electrical transduction and stereocilia maintenance.

**Keywords** hair cells; mechanotransduction; mouse models; mutagenesis; stereocilia

**Subject Categories** Genetics, Gene Therapy & Genetic Disease; Molecular Biology of Disease

## Introduction

The process of hearing requires the transduction of sound wave-induced mechanical energy into neuronal signals. This process is achieved by the mechanosensitive inner ear hair cells located in the cochlea. These specialized sensory cells, named inner hair cells (IHCs) and outer hair cells (OHCs), have an array of actin-filled stereocilia protruding from their apical surface. Each hair cell stereocilia bundle is arranged as 3–4 rows in a highly ordered "staircase-like" structure, which is essential for function. Each taller stereocilium is connected to a shorter neighbour, in an adjacent row, by an extracellular tip link (Kazmierczak *et al*, 2007), with the upper end of the tip link extending from the side of a taller-row stereocilium to the tip of a shorter-row stereocilium, where it is tethered to the transduction channel complex. In response to sound-induced fluid movement within the inner ear, hair cell bundles are deflected towards the tallest stereocilia

1   Mammalian Genetics Unit, MRC Harwell Institute, Harwell, UK
2   Déficits Sensoriels Progressifs, Institut Pasteur, INSERM UMR-S 1120, Sorbonne Universités, Paris, France
3   Department of Biomedical Science, University of Sheffield, Sheffield, UK
4   Department of Twin Research & Genetic Epidemiology, King's College London, London, UK
5   Génétique et Physiologie de l'Audition, Institut Pasteur, INSERM UMR-S 1120, Collège de France, Sorbonne Universités, Paris, France
6   Mary Lyon Centre, MRC Harwell Institute, Harwell, UK
7   Laboratoire de Neurophysiologie de la Synapse Auditive, Université de Bordeaux, Bordeaux, France
8   Department of Otolaryngology – Head and Neck Surgery, University Hospitals Cleveland Medical Center, Case Western Reserve University, Cleveland, OH, USA
9   Department of Otorhinolaryngology Head and Neck Surgery, Anatomy and Neurobiology and Institute for Genome Sciences, University of Maryland School of Medicine, Baltimore, MD, USA
10  UCL Ear Institute, University College London, London, UK
    *Corresponding author. Tel: +33 1 45 68 88 92; E-mail: aziz.el-amraoui@pasteur.fr
    **Corresponding author. Tel: +44 1235 841161; E-mail: m.bowl@har.mrc.ac.uk
    †These authors contributed equally to this work

causing tension in the tip links, which opens the mechanically gated transduction channels, allowing the influx of $K^+$ and $Ca^{2+}$ ions into the hair cell, leading to depolarization and release of neurotransmitter (Corey & Hudspeth, 1983; Schwander et al, 2010). Components of the elusive transduction channel complex include LHFPL tetraspan subfamily member 5 (LHFPL5), transmembrane inner ear (TMIE) and transmembrane channel-like 1 (TMC1) and TMC2 (Kawashima et al, 2011; Kurima et al, 2015; Corns et al, 2016, 2017; Fettiplace, 2016; Beurg et al, 2018). All these proteins are reported to interact with protocadherin-15 (PCDH15), a component of the tip link, anchoring it to the stereocilia membrane (Xiong et al, 2012; Maeda et al, 2014; Zhao et al, 2014). The development and maintenance of the "staircase" stereocilia bundle, and the inter-stereociliary tip links, are therefore critical for auditory transduction and essential for hearing. Currently, our knowledge of the mechanisms underlying stereocilia bundle formation and maintenance is limited, and the precise molecular composition of the transduction channel complex remains elusive.

The Clarin (CLRN) proteins belong to a superfamily of small integral proteins with four alpha-helical transmembrane domains, which also includes Tetraspanins, Connexins, Claudins, Occludins and calcium channel gamma subunit-like proteins (Adato et al, 2002; Aarnisalo et al, 2007). In humans, the CLRN family comprises three proteins encoded by the paralogous genes CLRN1, CLRN2 and CLRN3, which contain no known functional domains apart from their four transmembrane domains and a C-terminal class-II PDZ-binding motif (PBM type II) (Fig 1A). In humans, CLRN1 mutations have been found to cause Usher syndrome type 3A (USH3A), which is characterized by postlingual, progressive hearing loss, variable vestibular dysfunction and onset of retinitis pigmentosa leading to vision loss (Adato et al, 2002; Bonnet & El-Amraoui, 2012). Similarly, Clrn1 knockout ($Clrn1^{-/-}$) mice are reported to show early-onset profound hearing loss, and consistent with this, these mice exhibit disrupted stereocilia bundles in the early postnatal period (Geng et al, 2009, 2012). However, to date Clrn2 has not been associated with any disease and has never been the focus of a scientific paper.

Utilizing an unbiased forward genetic screen, we have identified an ENU-induced Clrn2 mutation as the cause of deafness in the clarinet mouse mutant ($Clrn2^{clarinet}$). Moreover, we have employed $Clrn2^{clarinet}$ mice and a second CRISPR/Cas9-induced mutant ($Clrn2^{del629}$) to investigate the requirement of clarin-2 in the auditory, vestibular and visual systems. While clarin-2 appears to have a nonessential role in the retina and vestibular apparatus, its absence leads to an early-onset progressive hearing loss. In addition, we identify that genetic variation at the human CLRN2 locus is highly associated with adult hearing difficulty in the UK Biobank Cohort. Expression of tagged clarin-2 in cochlear cultures shows enrichment of the protein in hair cell stereocilia. We demonstrate that clarin-2 is not required for the initial patterning, or formation, of the "staircase" stereocilia bundle, but instead is essential for the process of maintenance of the stereocilia bundle and mechano-electrical transduction. This study establishes a critical role for the tetraspan protein clarin-2 in the function of the mammalian auditory system.

# Results

## Clarin-2, a novel protein essential for mammalian hearing

During a recent phenotype-driven ENU-mutagenesis screen undertaken at the MRC Harwell Institute, pedigree MPC169 was identified as containing mice with hearing impairment (Potter et al, 2016). In a $G_3$ cohort of 69 mice, 8 were found to have severely elevated auditory brainstem response (ABR) thresholds at 3 months of age (Fig 1A). A genome scan and subsequent single nucleotide polymorphism (SNP) mapping of affected (deaf) and unaffected (hearing) $G_3$ mice demonstrated linkage to a ~12-Mb region on Chromosome 5 (Fig EV1A). Whole-genome sequencing of an affected mouse identified a homozygous mutation within the critical interval consisting of a non-synonymous G-to-A transition at nucleotide 12 of the Clrn2 gene (ENSMUST00000053250). The Clrn2 mutation, confirmed using Sanger sequencing (Fig EV1B), leads to a tryptophan-to-stop (p.Trp4*) nonsense mutation in the encoded clarin-2 protein, a tetraspan-like glycoprotein with a class-II PDZ-binding motif (Fig 1B and C). We subsequently named this mutant clarinet and backcrossed the $Clrn2^{clarinet}$ allele to C57BL/6J for ten generations.

To confirm $Clrn2^{clarinet}$ is the causal mutation underlying the auditory dysfunction observed in clarinet mice, we first used a CRISPR/Cas9 approach to engineer a second Clrn2 mutant mouse model, named $Clrn2^{del629}$. This allele consists of a 629 nucleotide deletion that encompasses exon 2 (ENSMUSE00000401986) of the Clrn2 gene, which encodes the second, and part of the third, transmembrane domains of clarin-2. As such, while the remaining exons 1 and 3 splice together and are in-frame, any translated protein is predicted to have reduced, or absent, function (Figs 1D and EV1C). Next, we undertook a complementation test crossing together these two Clrn2 mutant lines (Figs 1E and EV1D). ABR measurements, recorded in postnatal day 28 (P28) mice in response to click and tone-burst stimuli, showed that compound heterozygous ($Clrn2^{clarinet/del629}$) mice display very elevated thresholds (> 80 decibel sound pressure level (dB SPL)) at all frequencies tested: 8, 16 and 32 kHz, whereas $Clrn2^{clarinet/+}$ and $Clrn2^{del629/+}$ mice exhibit thresholds comparable with those of wild-type ($Clrn2^{+/+}$) littermates (< 40 dB SPL) (Figs 1E and EV1D), demonstrating the absence of a heterozygous auditory phenotype. Failure of complementation in $Clrn2^{clarinet/del629}$ mice confirms the gene Clrn2 is essential for hearing.

Utilizing the UK Biobank Cohort (Sudlow et al, 2015), a multiphenotype study of 500,000 people aged between 40 and 69 years, we also sought whether genetic variation at the CLRN2 locus is related to self-reported human hearing difficulty. The association was performed using a case–control design ($n = 250,389$) based on answers to questions regarding participants' self-assessed hearing ability and self-reported hearing difficulty in the presence of background noise. An association was tested between all 484 UK Biobank genotyped and imputed SNPs within 100 kb of the CLRN2 gene. Within this region, 36 SNPs were significantly associated with the hearing difficulty phenotype, including a cluster of five highly associated SNPs that lie within or very close to the CLRN2 gene (Fig 1F). Within the 20 most highly associated SNPs, the majority are either intronic or intergenic (Table EV1). The rs35414371 SNP with the highest association has a P-value of 1.60E-11 and lies just 2 kb downstream of the CLRN2 gene. The second most associated

**A**   ABR responses in a new deaf mutant mouse, *clarinet*, and control littermates at 3 months of age

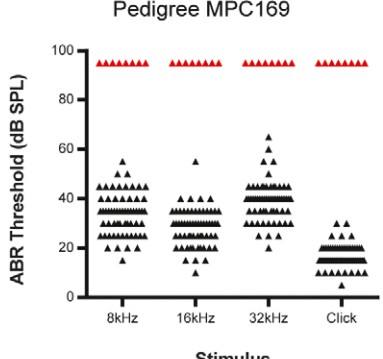

**B-E** Clarin-2 in wild-type and deaf mutant mice

**B**   mouse wild-type protein and *Clrn2* allele

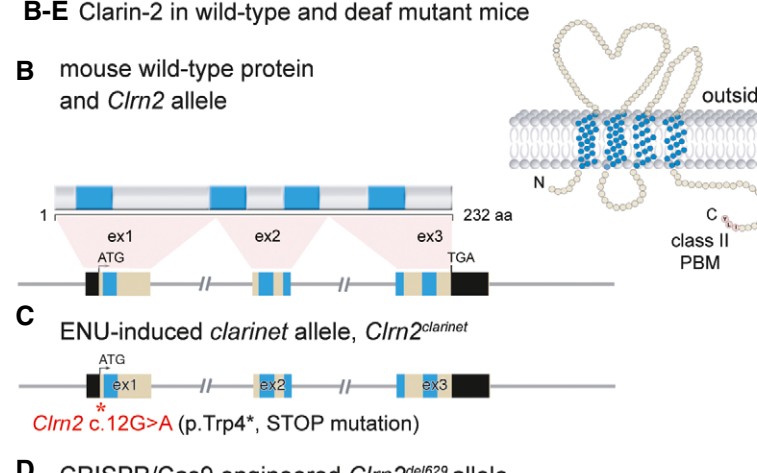

**C**   ENU-induced *clarinet* allele, *Clrn2^clarinet*

*Clrn2* c.12G>A (p.Trp4*, STOP mutation)

**D**   CRISPR/Cas9 engineered *Clrn2^del629* allele

deletion of a 629 bp region containing exon 2

**E**   ABR responses in *Clrn2* compound heterozygous P21 mice

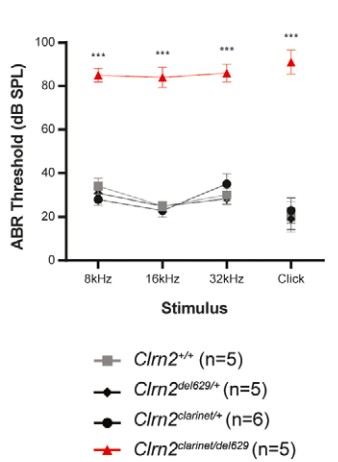

**F**

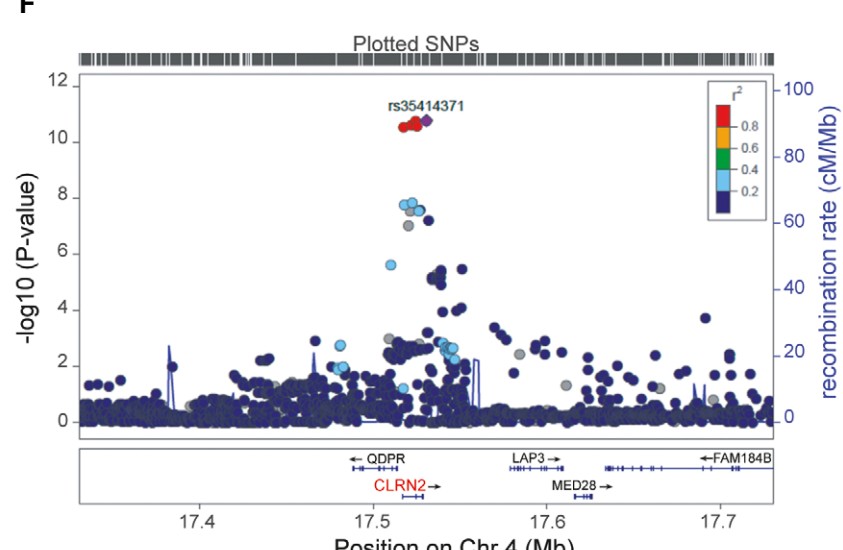

**Figure 1.   *Clrn2* is essential for mammalian hearing.**

A    Identification of the ENU-induced hearing loss pedigree MPC169, subsequently named *clarinet*. ABR phenotyping of pedigree Muta-Ped-C3PDE-169 at 3 months of age identified 8 mice with elevated hearing thresholds (red triangles) compared to their normal hearing colony mates (*n* = 61, black triangles). Indeed, all eight affected mice were found to not respond to the highest intensity stimulus (90 dB SPL) at the three frequencies tested, or the click stimulus, and so their thresholds are shown as 95 dB SPL.

B–D   The genomic structure of mouse *Clrn2* (ENSMUST00000053250), and domains of the encoded tetraspan-like glycoprotein (232 amino acids). Black-filled boxes represent untranslated region of *Clrn2*. The positions of the transmembrane (TM) domains (blue) and the structure of *Clrn2^clarinet* (C) and *Clrn2^del629* (D) alleles are indicated. The *clarinet* mutation, *Clrn2^clarinet* (c.12G > A) (red asterisk), is predicted to lead to a premature stop codon at position 4 (p.Trp4*) (C), whereas the *Clrn2^del629* allele consists of a CRISPR/Cas9-mediated 629 nucleotide deletion encompassing exon 2, leading to splicing of exon 1 to exon 3, which if translated would produce a protein lacking 2 (TM2 and TM3) of the 4 transmembrane domains (D).

E    Averaged ABR thresholds for *Clrn2^clarinet/del629* compound heterozygotes at P21, showing significantly elevated thresholds compared to *Clrn2^+/+*, *Clrn2^clarinet/+* and *Clrn2^del629/+* control colony mates. All five *Clrn2^clarinet/del629* mice were found to not respond at the highest intensity stimulus (90 dB SPL) for at least one frequency/ click stimulus. Data shown are mean ± SD ***$P < 0.001$, one-way ANOVA (Please see Appendix Table S1 for exact *P*-values).

F    Regional plot of *P*-values for SNP association with hearing difficulty around the *CLRN2* gene locus. The genes within the region are annotated, and the direction of the transcripts is shown by arrows. Colouring is based on linkage disequilibrium (LD) across the region with the most associated SNP, rs35414371, shown in purple.

SNP, rs13147559 ($P = 1.70E-11$), is in exon 2 of the *CLRN2* gene at coding nucleotide position 337 (c.337, ENST00000511148.2). Presence of the ancestral allele (cytosine, c.337C) encodes for leucine (p.113Leu), whereas presence of the minor allele (guanine, c.337G) encodes for valine (p.113Val). As such, this SNP (c.337C > G) represents a missense variant (p.Leu113Val) within the predicted

transmembrane domain 2 of the clarin-2 protein (NP_001073296). *In silico* studies show that the leucine at position 113 is evolutionarily conserved across species. Furthermore, two prediction tools, PolyPhen-2 and MutationAssessor, suggest that substitution of a valine at this position might be detrimental to clarin-2 function returning scores of "possibly damaging" and "medium", respectively.

Together, our findings indicate that clarin-2 is key to hearing in both mice and humans.

## Clarin-2 is essential for hearing function

Clarin-2 displays 56% amino acid similarity with clarin-1, the USH3A protein. Indeed, *CLRN1* loss of function has been shown to cause progressive hearing loss, variable vestibular dysfunction and progressive retinitis pigmentosa, which prompted us to seek whether *Clrn2* is a candidate Usher gene. RT–PCR analyses from wild-type P30 mice revealed the presence of *Clrn2* transcripts in the inner ear (notably in the auditory hair cells) and the eye, but not in brain or muscle (Fig 2A and B). Thus, functional measurements were performed to characterize hearing, vestibular and visual phenotypes in *clarinet* mice.

To establish the onset and progression of auditory impairment in $Clrn2^{clarinet/clarinet}$ mice, ABR measurements were undertaken at P16, which is just after the onset of hearing in mice (~P12), and longitudinally at P21, P28 and P42. At P16, $Clrn2^{clarinet/clarinet}$ mice display elevated hearing thresholds (e.g. mean click threshold 64 dB SPL ± 15 SD) compared with their littermate controls (mean click threshold < 30 dB SPL ± 6 SD for $Clrn2^{+/+}$ mice) (Fig 2C). At P21, $Clrn2^{clarinet/clarinet}$ mice display increased auditory thresholds (mean click threshold 80 dB SPL ± 16 SD) compared with P16 $Clrn2^{clarinet/clarinet}$ mice, and thresholds continue to increase by P28, and P42 (mean click threshold 89 dB SPL ± 5 SD; Fig 2D–F). To further assess cochlear function, distortion product otoacoustic emissions (DPOAEs) were measured in P28 $Clrn2^{clarinet/clarinet}$ mice. Compared to their $Clrn2^{+/+}$ and $Clrn2^{clarinet/+}$ littermates, $Clrn2^{clarinet/clarinet}$ mice have reduced DPOAEs (Fig 2G) suggesting impaired OHC function. These results show that lack of clarin-2 causes an early-onset hearing loss, characterized by a fast-progressive deterioration of hearing function likely affecting both inner and outer hair cells.

To assess for potential vestibular deficits, *clarinet* and control mice were subject to various tests, including platform, trunk-curl, contact righting and swim tests (Hardisty-Hughes *et al*, 2010). Regardless of the test employed, no overt difference between $Clrn2^{clarinet/clarinet}$ mice ($n = 12$) and age-matched $Clrn2^{clarinet/+}$ control mice ($n = 10$) was observed at P28 or at P60, indicating normal balance function in young animals despite the absence of clarin-2 (Fig 3A; $P > 0.05$).

To investigate for possible visual deficits in *clarinet* mice, we measured the retinal-evoked potential responses, characterized by an initial negative deflection (the a-wave) followed by a positive peak (the b-wave), the amplitudes of which vary with light intensity. Functional electroretinogram (ERG) measurements, under scotopic (Fig 3B, D and E) or photopic (Fig 3C and F) conditions, indicated that rod and cone functions are normal in $Clrn2^{clarinet/clarinet}$ mutant mice at both 3 months and 6–7 months. The ERG responses were almost normal in shape, with unaffected time-to-peak values for the

a- and b-waves (Fig 3B and C). The amplitudes measured at the peak of both the a- and b-waves were also similar for control ($Clrn2^{clarinet/+}$) and $Clrn2^{clarinet/clarinet}$ mice at 6–7 months (a-wave, Fig 3D: $172 \pm 14$ and $207 \pm 17$, respectively; b-wave, Fig 3E: $299 \pm 15$ and $242 \pm 39$, respectively; photopic ERGs, Fig 3F: $61 \pm 7$ and $67 \pm 8$, respectively). Consistent with ERG findings, the overall laminar organization of the retina in control $Clrn2^{clarinet/+}$ and mutant $Clrn2^{clarinet/clarinet}$ mice, examined on cryosections from mice aged 6–7 months, is normal, with normal retinal pigment epithelium, clearly distinguishable neuroretinal layers, and normal targeting and restriction of the short wavelength-sensitive opsin 1 (S opsin) and rhodopsin to the outer segment in blue cone and rod photoreceptor cells, respectively (Fig EV2A and B). No pyknotic nuclei, indicative of degenerating cells, were observed in any of the retinal cell layers, and TUNEL assays detected no apoptosis (Fig EV2C). The Iba1-immunoreactive microglial cells in the retinas of $Clrn2^{clarinet/clarinet}$ mice had features typical of the resting state similar to age-matched controls, including long thin neurites and lack of immunostaining in the photoreceptor cell-containing layer (Fig EV2D).

Together, our findings indicate that the absence of clarin-2 leads to an early-onset, progressive hearing loss, without overt retinal deficits.

## Clarin-2 is not required for the formation and proper polarization of the hair bundle

To investigate the cause of hearing loss in $Clrn2^{clarinet/clarinet}$ mice, and considering the previously reported disrupted organization of auditory hair bundles in neonatal *Clrn1* mutant mice (Geller *et al*, 2009; Geng *et al*, 2009, 2012; Dulon *et al*, 2018), we used confocal and scanning electron microscopy to monitor the progression of hair cell stereocilia bundle development and maturation from birth (Figs 4 and 5).

At P6, towards the end of the first postnatal week, $Clrn2^{clarinet/clarinet}$ cochlear OHC and IHC hair bundles have a cohesive appearance, with the expected V- and U-shape organization, respectively, compared to heterozygous controls (Fig 4A and B). Moreover, throughout $Clrn2^{clarinet/clarinet}$ cochleae, stereocilin localizes as expected at the tips of the tallest stereocilia (Fig 4C and E, $n = 3$), indicating normal coupling between OHC stereocilia and the overlying tectorial membrane (Verpy *et al*, 2008). These data contrast with the grossly misshapen auditory OHC hair bundles exhibited by *Clrn1* mutant mice (Geller *et al*, 2009; Geng *et al*, 2009; Dulon *et al*, 2018) (see also Fig 4D). By P8, scanning electron microscopy shows $Clrn2^{clarinet/clarinet}$ mutants do not exhibit any gross patterning defects, or differences in the overall number of OHC and IHC bundles compared to controls (Fig 4F and G). Mutant hair bundles throughout the cochlea still displayed the 3–4 rows of stereocilia arranged in a regular "staircase-like" pattern as in age-matched control mice (see Fig 5A and B). Also, cochlear OHC apical circumferences have lost their immature rounded shape, to acquire a non-convex form moulded to the V-shape of the overlying hair bundle (arrowheads in Fig 4A, B, F and G).

The first notable morphological defect in $Clrn2^{clarinet/clarinet}$ mice was observed in the auditory hair cells of the cochlea towards the end of the first postnatal week. Detailed analyses at P8 revealed that the shortest row stereocilia in mutant OHCs appear shorter than those of $Clrn2^{+/+}$ littermates (Fig 5A and C). In addition, the tips of

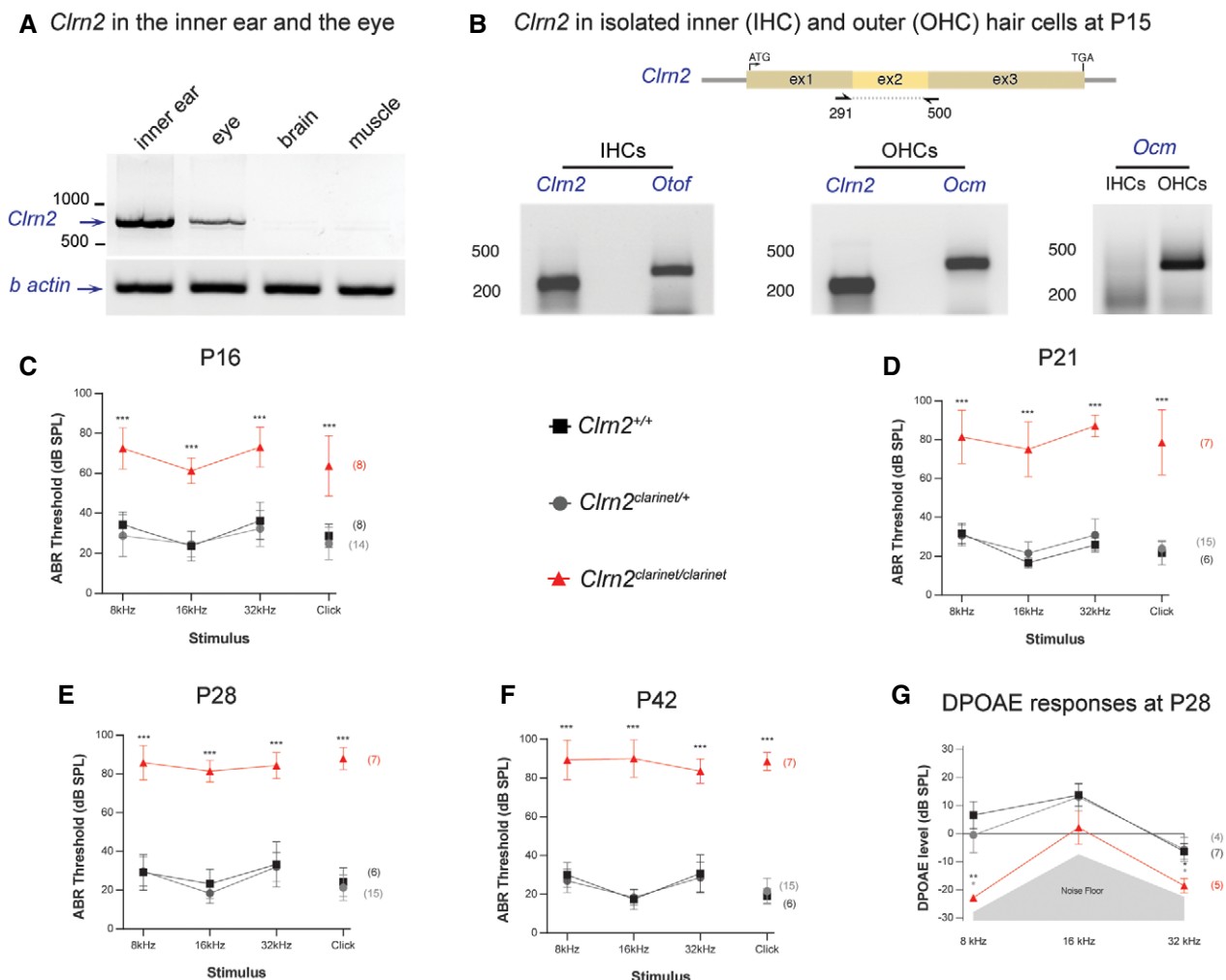

**Figure 2. Clarin-2 is required for hearing function.**

A    RT–PCR analysis in P30 mice showing the presence of *Clrn2* transcripts in the inner ear and eye, but not in brain or muscle. β-actin was used as a positive control.

B    *Clrn2* transcripts could be detected in both inner (IHCs) and outer (OHCs) hair cells of P15 wild-type mice. Otoferlin (*Otof*) and oncomodulin (*Ocm*) transcripts were used as positive controls for IHCs and OHCs, respectively. *Ocm* transcripts were only present in the OHC lysate, demonstrating that the IHC sample had not been contaminated with OHCs.

C–F    Auditory phenotyping of *clarinet* mice at P16 (C), P21 (D), P28 (E) and P42 (F). ABR threshold measurements show that $Clrn2^{clarinet/clarinet}$ mice (red) exhibit a severe-to-profound hearing loss affecting all frequencies tested. At 16 kHz in $Clrn2^{clarinet/clarinet}$ mice, the mean ABR hearing thresholds vary from 55–65 dB SPL at P16, 60–90 dB SPL at P21 and to 80–100 dB SPL at P28 and P42. Age-matched $Clrn2^{+/+}$ (black) and $Clrn2^{clarinet/+}$ (grey) controls display thresholds within the expected range (15–45 dB SPL) at all frequencies and timepoints tested. At P16, all eight $Clrn2^{clarinet/clarinet}$ mice exhibited recordable ABR responses for each frequency tested and the click stimulus. For the longitudinal ABR study, at P21 and P28 three of the seven $Clrn2^{clarinet/clarinet}$ mice were found to not respond at the highest intensity stimulus (90 dB SPL) for at least one frequency-specific/click stimulus. By P42, five of the $Clrn2^{clarinet/clarinet}$ mice were found to not respond at the highest intensity stimulus (90 dB SPL) for at least two frequency-specific/click stimuli. ABR data shown are mean ± SD ***P < 0.001, one-way ANOVA.

G    Averaged DPOAE responses for *clarinet* mice at P28, showing significantly reduced responses in $Clrn2^{clarinet/clarinet}$ mutants at all frequencies tested. DPOAE data shown are mean ± SD. *P < 0.02, **P < 0.01, one-way ANOVA. Please see Appendix Table S1 for exact P-values.

the middle and shortest row stereocilia in $Clrn2^{clarinet/clarinet}$ mutant IHCs are less prolate compared to $Clrn2^{+/+}$ littermates, instead displaying a rounded appearance (Fig 5B). Focusing on the second stereocilia row in both IHCs and OHCs, we used cochlear mid-turn electron micrographs from control and clarin-2-deficient hair bundles at P8 to score prolateness. At least 80 tip images per genotype were used—3 animals per genotype, 3 bundles per animal. We found a high prevalence of a rounded shape of the stereocilia from *clarinet* mice, as compared to age-matched wild-type mice, where

the normal prolate shape is far more common (P < 0.005 for all cases, χ²) (Fig EV3). To further characterize the bundle architecture, we measured the heights of OHC and IHC stereocilia (tallest, middle and shortest rows) within individual hair cell bundles from the mid-region of the cochlea (≥ 9 stereocilia per bundle, ≥ 2 bundles per animal, 3 animals per genotype). At P8 in OHC bundles, the average heights of stereocilia in all three rows are significantly shorter in $Clrn2^{clarinet/clarinet}$ mice relative to $Clrn2^{+/+}$ littermates (tallest −13%, middle −25%, shortest −52%) (Fig 5C). However, at P8 in

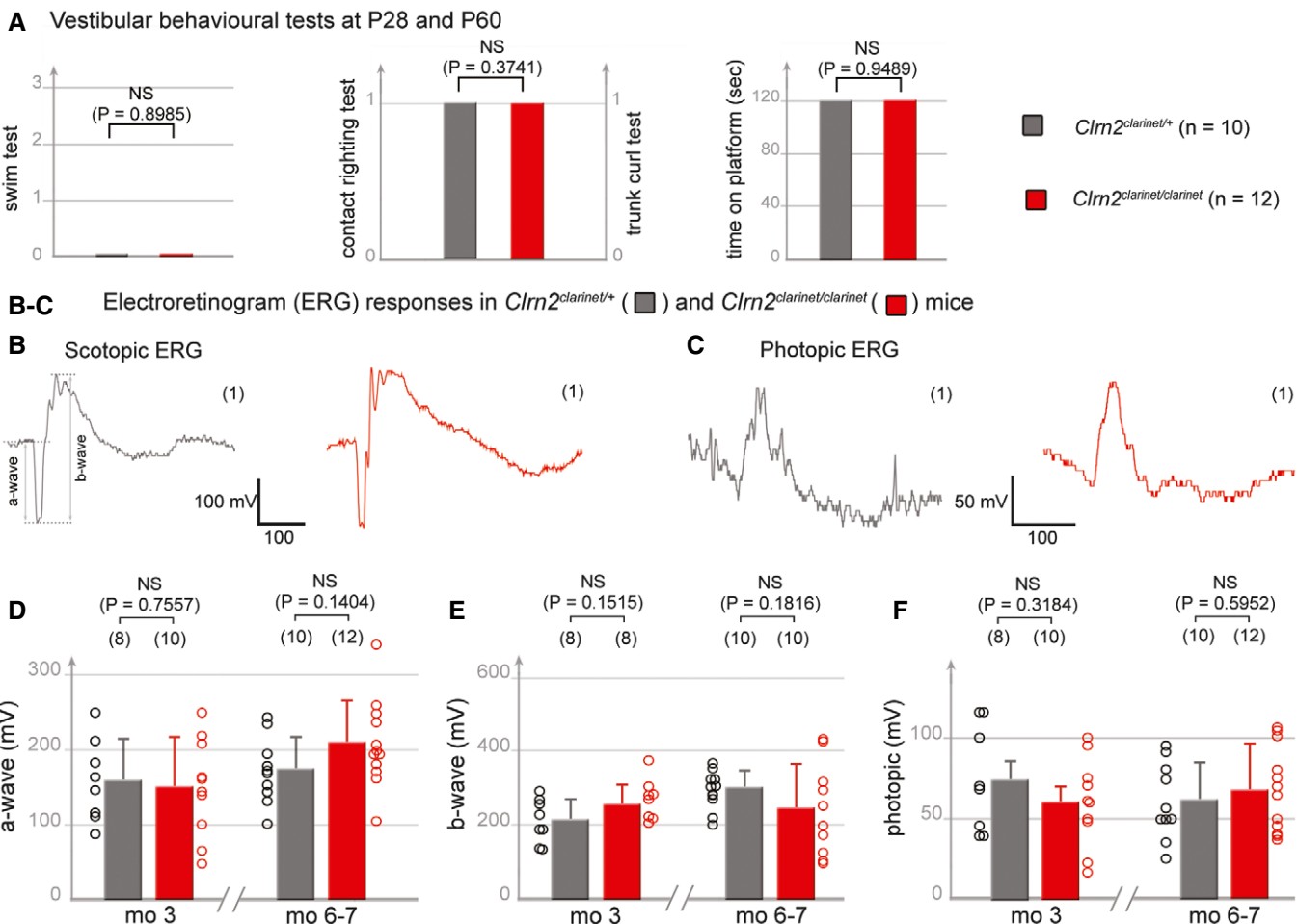

**Figure 3. Clarin-2 is dispensable for balance and vision.**

A   Vestibular behavioural tests (swim tests, contact-righting, trunk-curl and platform). The *Clrn2*$^{clarinet/clarinet}$ mice (red, P28, *n* = 5, and P60, *n* = 7) have no vestibular dysfunction, displaying similar performances to age-matched control *Clrn2*$^{clarinet/+}$ mice (grey, P28, *n* = 3, and P60, *n* = 7) (Student's *t*-test was used for the platform assay, and the Pearson's Chi-squared test for other experiments). Being similar, the values at P28 and P60 were combined.

B–F   Electroretinogram (ERG) measurements from control *Clrn2*$^{clarinet/+}$ (grey) and mutant *Clrn2*$^{clarinet/clarinet}$ (red) mice. Each trace in (B, C) is representative of an ERG response from the eye of age-matched *Clrn2*$^{clarinet/+}$ and *Clrn2*$^{clarinet/clarinet}$ mice, showing no significant difference in a- or b-wave amplitudes. (D–F) The lack of change in ERG amplitude responses in *Clrn2*$^{clarinet/clarinet}$ mice (aged 3 or 6–7 months), regardless of the test conditions: scotopic (D, E) or photopic (F) indicates normal photoreceptor kinetics and no change in the sensitivity of photoreceptor cells. The data shown are mean ± SEM. (NS) indicates a statistically non-significant difference (*P* > 0.1, Student's *t*-test).

IHC bundles we do not identify any significant differences in the average heights of stereocilia between *Clrn2*$^{clarinet/clarinet}$ mutants and *Clrn2*$^{+/+}$ controls in any row (Fig 5D). By P16 in OHC bundles, the average height of stereocilia in all three rows continue to be shorter in *Clrn2*$^{clarinet/clarinet}$ mice relative to *Clrn2*$^{+/+}$ littermates (tallest −9%, middle −25%, shortest −87%), with many short row stereocilia now missing (Fig 5C), and by P28, this row is entirely absent (Fig 5E, left panels). At P16 in IHC bundles, similar to P8 we do not identify any differences in the average heights of IHC stereocilia between *Clrn2*$^{clarinet/clarinet}$ mutants and *Clrn2*$^{+/+}$ controls in any row (Fig 5B and D). However, the tips of the middle and shortest row IHC stereocilia in *Clrn2*$^{clarinet/clarinet}$ mice continue to display a rounded appearance, unlike the pronounced prolate shape observed in *Clrn2*$^{+/+}$ littermates (Fig 5B). Furthermore, the height of the shorter-row stereocilia is more variable with some very short stereocilia measured (Fig 5D). Moreover, by P28 the heights of the

middle and short row are visibly more variable in *Clrn2*$^{clarinet/clarinet}$ mice, and missing short row IHC stereocilia are evident (Fig 5E, right panels).

These data suggest that while clarin-2 is dispensable for patterning and establishment of the "staircase" bundle in young postnatal hair cells, it is critical for the maintenance of the transducing stereocilia in functionally mature inner and outer hair cells.

### Clarin-2 is targeted to the hair bundles of the hair cells

To analyse the expression profile of *Clrn2*, whole-cochlea RNA extracts were prepared from wild-type mice at different embryonic and postnatal timepoints and utilized for quantitative RT–PCR (qRT–PCR) analysis of *Clrn2* transcripts. This shows that the relative abundance of *Clrn2* transcripts in the cochlea is stable from late embryonic stages to P12 (the onset of hearing in mice), but

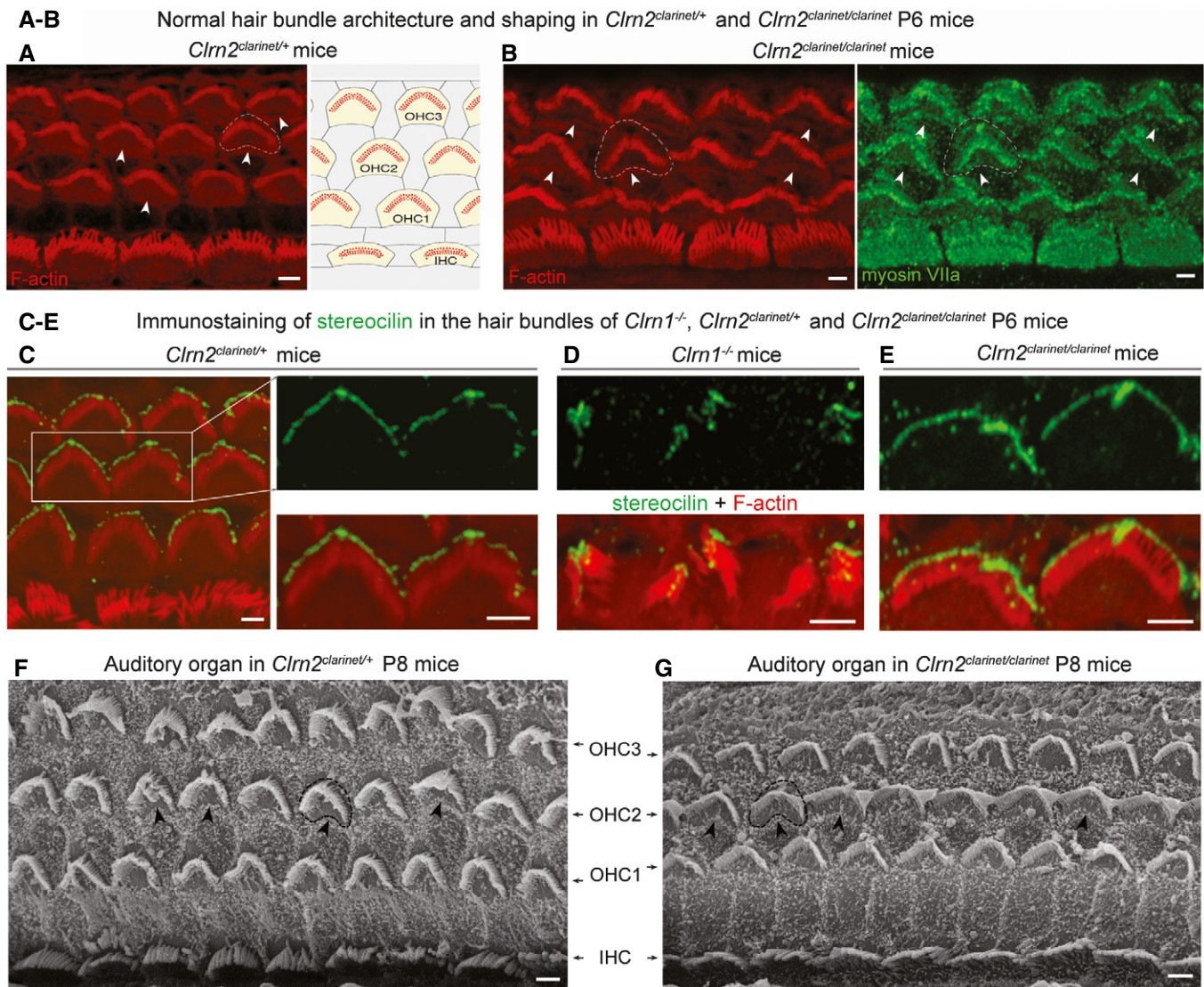

**Figure 4.  Normal architecture of cochlear hair bundles in *clarinet* mice.**

A, B  Confocal microscopy images of whole-mount preparations of mid-basal cochlear sensory epithelia from *Clrn2^clarinet/+* (A) and *Clrn2^clarinet/clarinet* (B) P6 mice immunostained for actin. Despite lack of clarin-2, the developing sensory epithelium of mutants is similar to that of heterozygous controls.

C–E  Confocal microscopy images of whole-mount preparations of cochlear sensory epithelia from *Clrn2^clarinet/+* (C), *Clrn1^−/−* (D) and *Clrn2^clarinet/clarinet* (E) P6 mice immunostained for stereocilin (green) and actin (red). Unlike the fragmented immunostaining in *Clrn1^−/−* mice (mirroring the fragmentation of the hair bundle), stereocilin immunostaining in *Clrn2^clarinet/clarinet* mice reflects the normally V-shaped bundles of OHCs, similar to *Clrn2^clarinet/+* OHCs.

F, G  Representative scanning electron micrographs of the sensory epithelium of *Clrn2^clarinet/+* (F) and *Clrn2^clarinet/clarinet* (G) P8 mice, showing no apparent differences in the gross patterning of IHCs and OHCs.

Data information: Arrowheads in (A, B) (white), (F, G) (black) illustrate the convex shape of the apical circumference of OHCs. Scale bars: 2 μm (A-E), 10 μm (F, G).

thereafter increases (Fig EV4A). Consistent with this finding, *in silico* analyses of the expression of *Clrn2* at different inner ear developmental and adult stages using the gEAR portal (*umgear.org*) reveal that while the *Clrn2* transcript is lowly expressed in the newborn inner ear and early postnatal stages, it is readily detected in P15 single-cell and adult-sorted hair cells (Liu *et al*, 2014, 2018; Ranum *et al*, 2019). In addition, the expression of *Clrn2* is detected in zebrafish hair cells (Steiner *et al*, 2014; Erickson & Nicolson, 2015). Interestingly, *Clrn2* transcripts were detected also in the auditory cortex (A1) and increased in levels

between P7 and adult mice (Guo *et al*, 2016). The *Clrn2* expression in both IHCs and OHCs was confirmed by RT–PCR on isolated auditory hair cells from P15 mice (Fig 2B). Regarding protein localization, several attempts were made to immunodetect endogenous mouse clarin-2, including raising an antibody against mouse clarin-2. While this purified antibody was able to recognize over-expressed clarin-2 in transfected cells, it does not detect endogenous clarin-2 despite various tests using cochlear whole mounts from different postnatal stages, and under various conditions of fixation and antigen-retrieval (Fig EV4B–E). Therefore, an

injectoporation approach (Xiong *et al*, 2014) was utilized to deliver a GFP-tagged clarin-2 expression construct to P2 cochlear cultures (see Fig 6A). After incubation, the injectoporated organs of Corti were fixed and co-stained with an anti-GFP antibody to detect the clarin-2 fusion protein, and phalloidin to visualize the stereocilia. In contrast to supporting cells, where the GFP-tagged clarin-2 is distributed diffusely throughout the cytoplasm (Fig 6B and C), in auditory hair cells, clarin-2 is enriched in the apical stereocilia (Fig 6D–F).

## Lack of clarin-2 causes selective defects in the mechano-electrical transduction machinery

To study molecular underpinnings of the bundle stereocilia remodelling in *clarinet* mice, we explored the distribution of myosin VIIa (used as a hair cell marker, Fig EV5A) and some selected proteins key to stereocilia growth and hair bundle organization (Figs 7 and EV5B–D). Considering the clarin-2 C-terminal class-II PDZ-binding motif, we investigated whether the absence of clarin-2 might

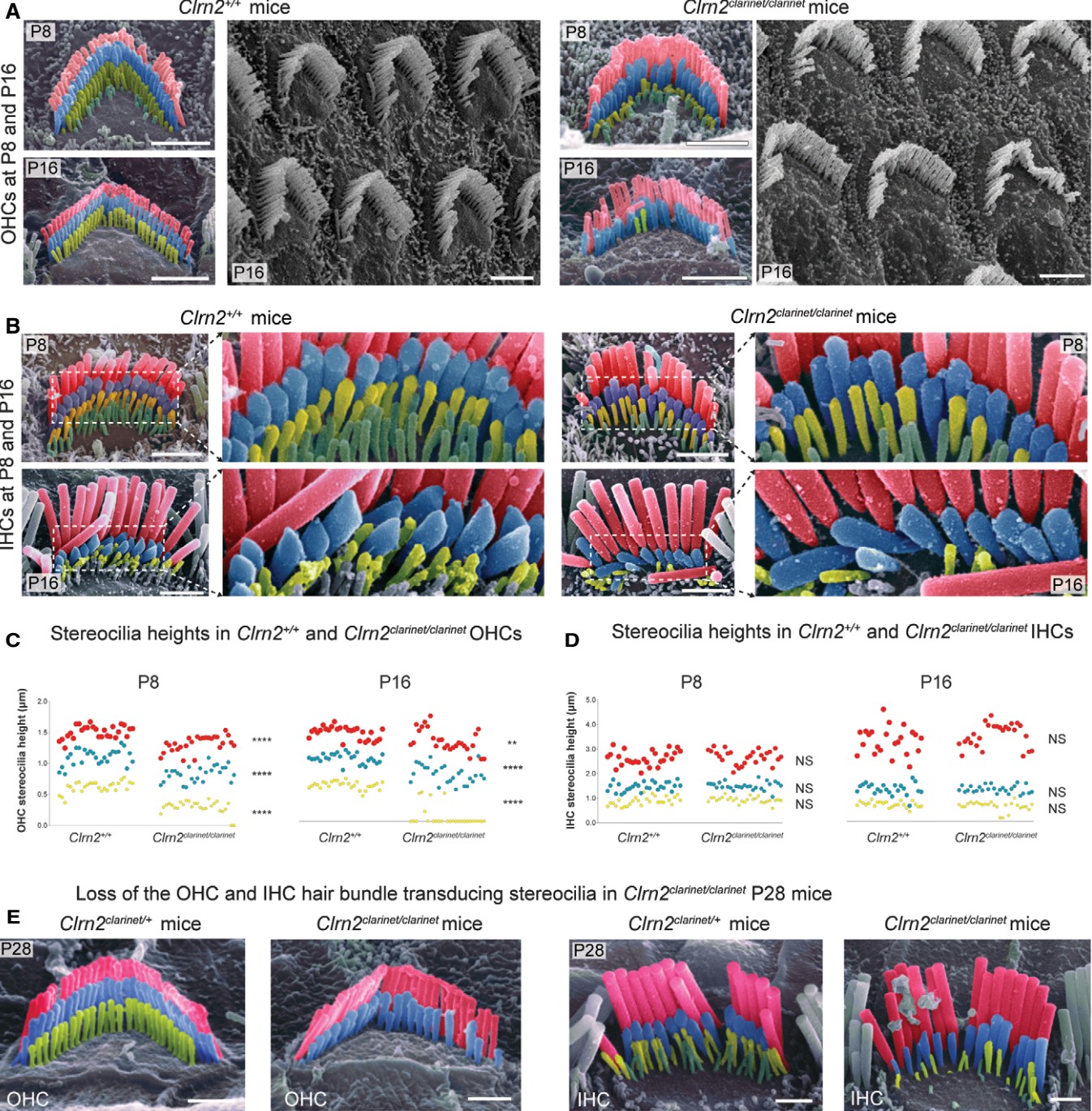

**Figure 5.**

**Figure 5. Sensory hair cell bundle patterning and measurements in *clarinet* mice.**

A, B  Pseudo-coloured scanning electron micrographs of individual outer and inner hair cell bundles from *clarinet* mice at P8 and P16, showing that gross morphology of OHC and IHC bundles is similar between $Clrn2^{+/+}$ and $Clrn2^{clarinet/clarinet}$ mice. Representative images from the mid-region of the cochlear spiral are shown, and close-up views illustrate the three full rows, tallest (red), middle (blue) and short (yellow), of stereocilia in IHC and OHC hair bundles. At P8 and P16 (upper panels in A and B), the three stereocilia rows are observed in $Clrn2^{clarinet/clarinet}$ OHCs (A) and IHCs (B), although the shortest and middle rows of IHC stereocilia appear less prolate compared to controls.

C, D  Distribution of individual stereocilia measures across genotypes at P8 and P16. (C) The OHC stereocilia measurements at both P8 and P16 (C) show a significant difference in height of $Clrn2^{+/+}$ stereocilia compared to $Clrn2^{clarinet/clarinet}$ in: e.g. for the shortest row at P8: $Clrn2^{+/+}$: 0.6188 ± 0.0944 p.µm (n = 27); $Clrn2^{clarinet/clarinet}$: 0.2948 ± 0.1029 p.µm (n = 27) (P < 0.0001, Mann–Whitney ranks comparison), and at P16: $Clrn2^{+/+}$: 0.579 ± 0.06179 p.µm (n = 27); $Clrn2^{clarinet/clarinet}$: 0.07282 ± 0.1626 p.µm (n = 27) (P < 0.0001, Mann–Whitney ranks comparison). Note the amount of value points equal to zero at P16. (D) Conversely, IHC stereocilia measurements at P8 and P16 did not show a difference between $Clrn2^{+/+}$ and $Clrn2^{clarinet/clarinet}$ in: for the shortest row at P8: $Clrn2^{+/+}$: 0.8808 ± 0.1753 p.µm (n = 27); $Clrn2^{clarinet/clarinet}$: 0.9313 ± 0.1217 p.µm (n = 27) (P = 0.2244, unpaired t-test); and at P16: $Clrn2^{+/+}$: 0.6784 ± 0.1171 p.µm (n = 23); $Clrn2^{clarinet/clarinet}$: 0.6063 ± 0.1855 p.µm (n = 26) (P = 0.2436, Mann–Whitney ranks comparison). **P < 0.01; ****P < 0.0001; NS P > 0.05.

E  Pseudo-coloured scanning electron micrographs of individual OHC (left panels) and IHC (right panels) hair bundles from *clarinet* mice at P28. $Clrn2^{clarinet/clarinet}$ mutants have only two rows of OHC stereocilia, and the middle row of stereocilia is less uniform in height compared to controls. The middle and short rows of stereocilia in $Clrn2^{clarinet/clarinet}$ IHC bundles appear fewer in number, and heterogeneous in height.

Data information: Scale bars, 2 µm.

interfere with the subcellular distribution of PDZ-containing proteins, namely whirlin, PDZD7 and harmonin. These deafness defective PDZ-containing adaptor proteins have been shown to anchor stereocilia integral membrane proteins to the underlying cytoskeleton (Boeda *et al*, 2002; Mburu *et al*, 2003; Grati *et al*, 2012). Within stereocilia, the subcellular distribution of whirlin and PDZD7 immunostainings were identical in $Clrn2^{clarinet/clarinet}$ mice and $Clrn2^{clarinet/+}$ littermates, consistent with normal growth and proper shaping of the auditory hair bundles, respectively (Figs 7A and B, and EV5B; n = 5). Interestingly, in $Clrn2^{clarinet/clarinet}$ mice, the PDZD7 immunoreactive spots were arranged mostly in one row, between middle and tallest stereocilia (arrow), while in age-matched $Clrn2^{clarinet/clarinet}$ control mice, PDZD7 was additionally detected between short and middle stereocilia (arrowhead) rows (Figs 7A and EV5B).

We also investigated the distribution of the harmonin-b isoform, a core component in mechano-electrical transducer (MET) transduction machinery (Grillet *et al*, 2009; Michalski *et al*, 2009), which has been shown to directly bind to actin filaments anchoring the apical-most tip link component, cadherin-23, to the stereocilia underlying cytoskeleton (Kazmierczak *et al*, 2007). In wild-type mice, between P1 and P5, harmonin-b localization switches from the stereocilia tips to a region below the tip of tall and medium stereocilia, corresponding to the upper attachment point of the tip link (Lefevre *et al*, 2008). This switch also occurs in the absence of clarin-2, since almost all $Clrn2^{clarinet/clarinet}$ stereocilia tips are devoid of harmonin-b staining (Fig 7C and D). Nevertheless, the harmonin-b immunoreactive spots were mispositioned; being located much closer to the stereocilia tip in $Clrn2^{clarinet/clarinet}$ mice compared to age-matched P6 controls (Figs 7C and D, and EV5C). We used confocal micrographs to quantify the positioning of harmonin-b immunoreactive spots in the stereocilia at the mid-basal region of the cochlea (Fig 7E). Measurements were performed using IHC, rather than OHC, bundles, focusing on the tallest stereocilia to allow accurate measurements of the distance of the harmonin-b immunoreactive spot from the stereocilium tip in $Clrn2^{clarinet/+}$ and $Clrn2^{clarinet/clarinet}$ P6 mice (Fig 7E). The positioning of the harmonin-b immunoreactive spots, relative to the tip of the tallest stereocilium, was observed on average at 575 ± 23 nm (mean ± SEM; n = 35 hair bundles from 5 mice) in $Clrn2^{clarinet/clarinet}$ mice, as compared to 850 ± 28 nm (mean ± SEM; n = 31 hair bundles

from 5 mice) in $Clrn2^{clarinet/+}$ mice (Fig 7F) (P < 0.0001, Student's t-test).

Given the reported function of clarin-1 in maturation of IHC ribbon synapses (Zallocchi *et al*, 2012; Ogun & Zallocchi, 2014), we also used immunolabelling to examine the ribbon synapse in *clarinet* cochlear whole mounts. In P21 apical-coil IHCs, a similar number of pre-synaptic ribbons (Ribeye-positive puncta) and post-synaptic densities (GluR2-positive puncta) were observed in $Clrn2^{+/+}$ and $Clrn2^{clarinet/+}$ control and $Clrn2^{clarinet/clarinet}$ mutant mice (Fig 7G and H). Furthermore, Ribeye-positive puncta and GluR2-positive puncta were juxtaposed in all genotypes (Fig 7G and H), as previously described (Brandt *et al*, 2005; Valeria *et al*, 2013), indicating normal coordination of synaptic body components and functional IHC ribbon synapses.

### Lack of clarin-2 disrupts hair cell transducer currents in auditory hair cells

Harmonin-b relocation to the upper link tip link density has been ascribed to its interaction with the tip link component cadherin-23 (Boeda *et al*, 2002; Grillet *et al*, 2009; Michalski *et al*, 2009). Harmonin-b mispositioning in the absence of clarin-2 (Fig 7D–F) might infer a decrease in the MET-induced tension forces, which is consistent with the high prevalence of round- and oblate-shaped stereocilia tips at this stage in $Clrn2^{clarinet/clarinet}$ mice (Figs 5B and EV3). To further investigate hair cell function, MET currents were recorded from P6–8 apical-coil OHCs by displacing their hair bundles in the excitatory and inhibitory direction using a piezo-driven fluid jet (Corns *et al*, 2014, 2016). At hyperpolarized membrane potentials (−121 mv), the displacement of the hair bundle in the excitatory direction (i.e. towards the taller stereocilia) elicited a large inward MET current in OHCs from both $Clrn2^{clarinet/+}$ and $Clrn2^{clarinet/clarinet}$ mice (Fig 8A and B). The maximal MET current in 1.3 mM $Ca^{2+}$ was significantly different (P < 0.0001) between $Clrn2^{clarinet/+}$ (−1,842 ± 66 pA at −121 mV, n = 7) and $Clrn2^{clarinet/clarinet}$ (−902 ± 39 pA, n = 9) OHCs (Fig 8C). The resting current flowing through open MET channels in the absence of mechanical stimulation was reduced when bundles were moved in the inhibitory direction (i.e. away from the taller stereocilia) in all OHCs tested (Fig 8A and B, arrows). Despite the different size of the maximal MET current, the open probability

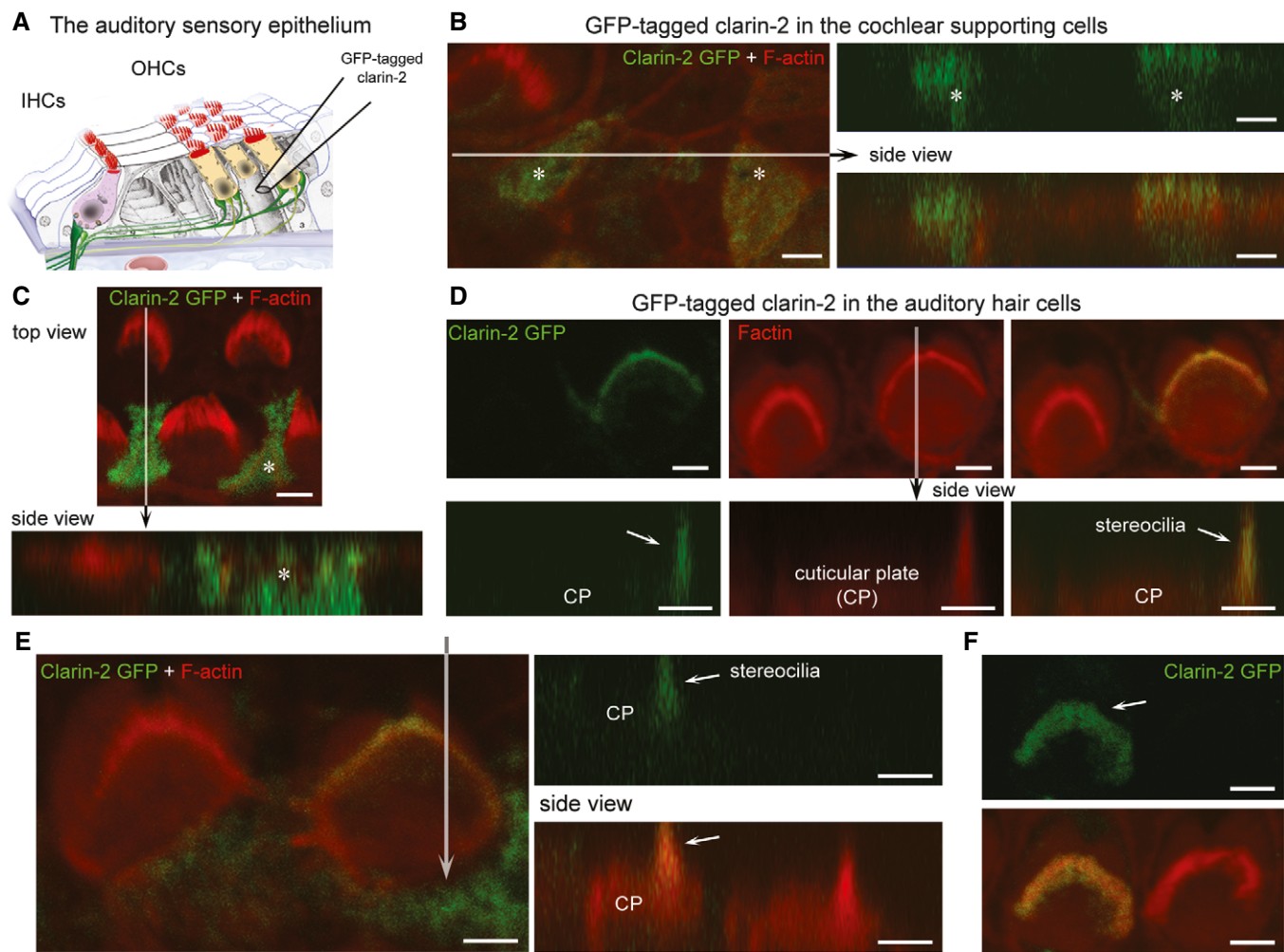

**Figure 6. Expression of clarin-2 in the hair bundle of auditory hair cells.**

A   Schematic representation of the auditory sensory organ, illustrating the positioning of the electrode used for injectoporation of GFP-tagged clarin-2 construct into cochlear supporting and hair cells.

B–F   Top and side views of representative images of supporting cells (asterisk) and hair cells expressing clarin-2. In supporting cells, GFP-tagged clarin-2 (green) was distributed diffusely throughout the cytoplasm (B, C). By contrast, in all injectoporated hair cells, the majority of clarin-2 was observed in the apical stereocilia (D–F). Arrows indicate position of the hair bundle stereocilia. Scale bars, 2 μm.

of MET channels was similar between $Clrn2^{clarinet/+}$ and $Clrn2^{clarinet/clarinet}$ OHCs (Fig 8D and E). Because the MET current reverses near 0 mV, it became outward when excitatory bundle stimulation was applied during voltage steps positive to its reversal potential (Fig 8A–C). At positive membrane potentials (+99 mV), the larger resting MET current (Fig 8A and B, arrowheads), which is due to an increased open probability of the transducer channel resulting from a reduced driving force for $Ca^{2+}$ influx (Crawford *et al*, 1989; Corns *et al*, 2014), was also similar between the two genotypes (Fig 8D). Similar findings were also observed in basal-coil OHCs (Fig 8E) and apical-coil IHCs (Fig 8F–H), with a significant reduction in the maximum MET current in $Clrn2^{clarinet/clarinet}$ mutants ($P < 0.0001$ for both hair cell types), but similar resting open probability, between $Clrn2^{clarinet/+}$ and $Clrn2^{clarinet/clarinet}$ mice.

Overall, these data show that clarin-2 is required for normal MET current in developing cochlear hair cells.

**Clarin-2 is required for the functional differentiation of IHCs, but not OHCs**

In the mouse cochlea, the onset of adult-like characteristics in OHCs occurs at around P8 with the expression of the negatively activated $K^+$ current $I_{K,n}$ (Marcotti & Kros, 1999) carried by the KCNQ4 channel (Kubisch *et al*, 1999). Potassium currents in apical-coil OHCs from P22 *clarinet* mice were elicited by applying a series of depolarizing voltage steps in 10 mV increments from −144 mV (holding potential was −84 mV). We found that adult OHCs from $Clrn2^{clarinet/+}$ and $Clrn2^{clarinet/clarinet}$ mice express the same complement of $K^+$ currents (Fig 9A–C). The size of the total outward $K^+$ current at 0 mV, which includes $I_{K,n}$ and the classical delayed rectified outward $K^+$ current $I_K$ (Marcotti & Kros, 1999), was similar between $Clrn2^{clarinet/+}$ (2.9 ± 0.6 nA, $n = 5$) and $Clrn2^{clarinet/clarinet}$ (3.2 ± 0.5 nA, $n = 6$) P22 OHCs. The size of $I_{K,n}$ was also similar

between the two genotypes. These results indicate that absence of clarin-2 does not influence the acquisition of the adult-like basolateral membrane properties of OHCs.

Different to the OHCs, the onset of adult-like characteristics in IHCs occurs at around P12 (Kros et al, 1998; Marcotti, 2012). The

IHC functional maturation is achieved by the down-regulation of immature-type currents (e.g. $Na^+$ current and the small conductance $Ca^{2+}$-activated $K^+$ current carried by SK2 channels) and inward rectifier (K1) currents) and expression of $I_{K,n}$ and the fast-activating large-conductance $Ca^{2+}$-activated $K^+$ current ($I_{K,f}$: carried by BK

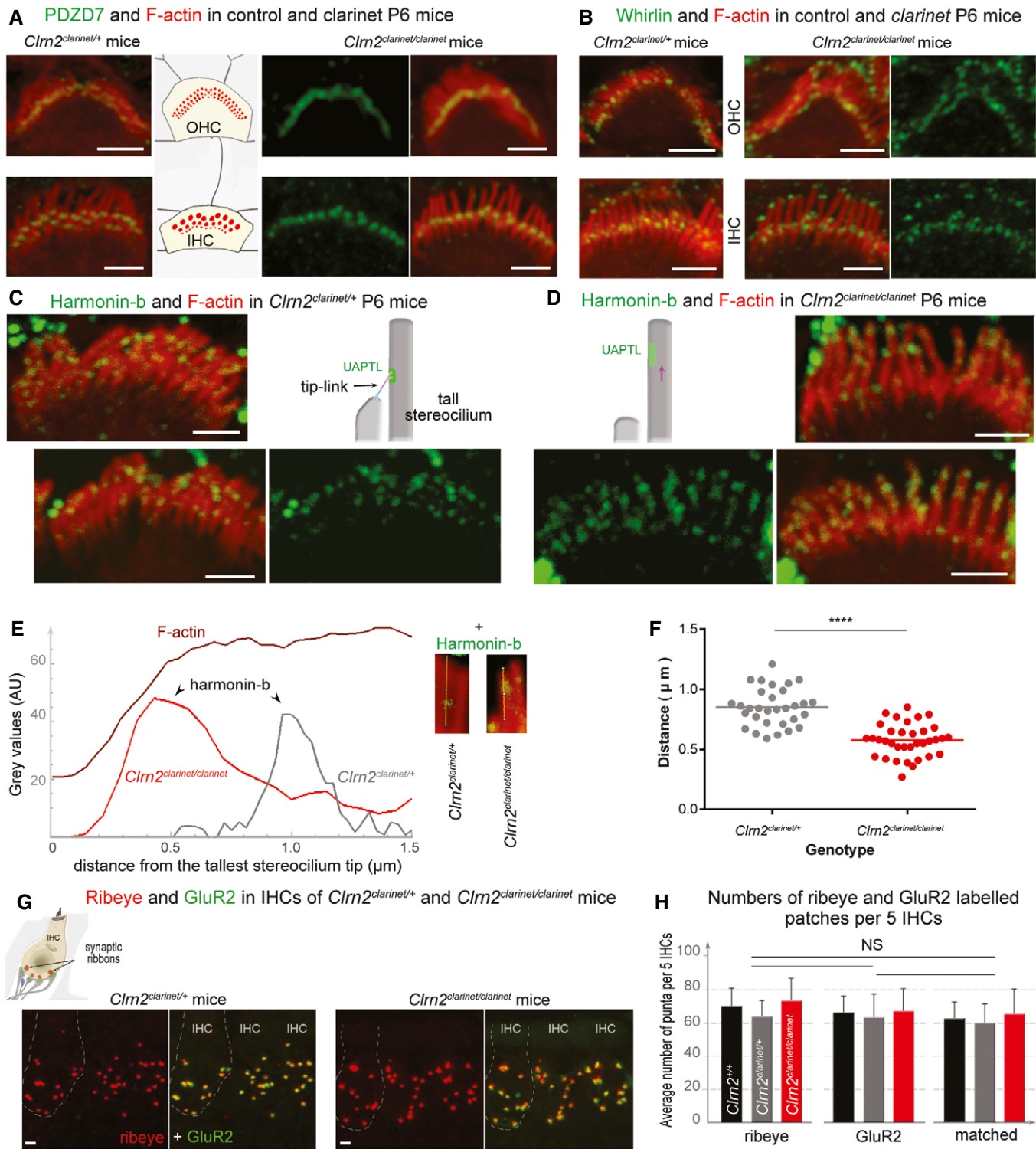

**Figure 7.**

**Figure 7. The distribution of hair bundle and synaptic proteins in *clarinet* mice.**

A, B   Confocal images of IHC and OHC hair bundles of *Clrn2$^{clarinet/clarinet}$* mice and heterozygous littermates at P6 immunostained for PDZD7 (A, green) and whirlin (B, green) and actin (red in both figures). The PDZD7 immunostaining is normally restricted to the base of stereocilia in both *Clrn2$^{clarinet/+}$* and *Clrn2$^{clarinet/clarinet}$* P6 mice (A). Whirlin immunostaining is properly located at the stereocilia tips of IHC and OHC hair bundles (B).

C–F   Harmonin-b immunostaining in IHC hair bundles. In *Clrn2$^{clarinet/clarinet}$* mice (D), the harmonin-b immunoreactive puncta (green) were still observable on the stereocilia, but unlike in age-matched heterozygote littermates (C), were located much closer to the tip of stereocilia (indicated by the purple arrow). The two diagrams in (C) and (D) illustrate the position of harmonin-b immunostaining (green) corresponding to the site of the upper attachment point of the tip link (UAPTL), facing the tip link. The bright green signal outside stereociliary bundles are non-specific. The change of harmonin-b localization along the stereocilium in *Clrn2$^{clarinet/clarinet}$* mice is illustrated further by line scan (E) and quantification (F) analyses. The insets in (E) show images of individual stereocilia used for the line scan signal analysis. The harmonin-b immunoreactive puncta were located within 850 ± 28 nm (mean ± SEM) distance from the tip of the tallest stereocilia in *Clrn2$^{clarinet/+}$* mice (n = 31 hair bundles from 5 mice) (F), and within 575 ± 23 nm in *Clrn2$^{clarinet/clarinet}$* mice (n = 35 hair bundles from 5 mice). Individual data points are shown and mean indicated by a horizontal bar. ****$P < 0.0001$, Student's $t$-test.

G, H   (G) Cochlear whole mounts from P21 *clarinet* mice, labelled with the IHC pre-synaptic ribbon marker Ribeye (red) and the post-synaptic density marker GluR2 (green), showing a similar number of total and matched Ribeye-positive and GluR2-positive puncta in *Clrn2$^{clarinet/clarinet}$* mutant cochleae compared to *Clrn2$^{clarinet/+}$* littermates, which is reflected in puncta counted per five hair cells (H) n = 4 per genotype. Data shown are mean ± SD, one-way ANOVA.

Data information: Scale bars, 2 μm.

channels) (Marcotti *et al*, 2003). Initially, we measured the size of the total outward K$^+$ current at 0 mV in P22 IHCs and found that it was similar between *Clrn2$^{clarinet/+}$* controls (11.6 ± 1.6 nA, $n = 4$) and *Clrn2$^{clarinet/clarinet}$* mutants (10.7 ± 1.0 nA, $n = 5$). For these experiments, currents were elicited by applying depolarizing voltage steps in 10 mV nominal increments from –124 mV up to + 30 mV, starting from the holding potential of −84 mV. However, a close inspection of the time-course of current activation showed some differences, which were evaluated in more detail by delivering a voltage protocol that allowed the evaluation of both $I_{K,n}$ and $I_{K,f}$ (Fig 9D–F). IHCs were held at −64 mV and subjected to depolarizing voltages in 10 mV nominal increments from −144 mV to more positive values. We found that while *Clrn2$^{clarinet/+}$* IHCs exhibit both $I_{K,n}$ and $I_{K,f}$, *Clrn2$^{clarinet/clarinet}$* IHCs fail to show the above currents and instead express $I_{K1}$ (Fig 9D–F), which is characteristic of a pre-hearing IHC (Marcotti *et al*, 1999). The physiological consequence of the above abnormalities was that adult *Clrn2$^{clarinet/clarinet}$* IHCs did not acquire the fast-graded voltage responses to stimulation normally present in mature cells (Fig 9G and H) and, in some cases, retained the ability to fire an initial Ca$^{2+}$-dependent action potentials (Fig 9I) that is characteristic of immature cells (Kros *et al*, 1998; Marcotti *et al*, 2003). We found that immature-type currents were not down-regulated in *Clrn2$^{clarinet/clarinet}$* IHCs, and as such, their functional differentiation into mature cells was prevented.

## Discussion

We demonstrate that *Clrn2* is a novel deafness gene required for maintenance of transducing stereocilia in the sensory cochlear hair cells. Our results show that the absence of clarin-2 leads to an early-onset hearing loss in *clarinet* mice, which is moderate-to-severe at P16, but rapidly progresses to profound hearing loss after P21. Moreover, our morpho-functional and molecular studies of clarin-2-deficient mice demonstrate this protein is critically required for late-stage cochlear hair bundle maintenance and function. In particular, we show that while clarin-2 is dispensable for the acquisition of polarized and assembled stereocilia bundles, the protein is essential for maintaining bundle integrity and normal sound-induced mechano-electrical transduction.

Hair bundle formation and polarization is a multi-step process with three main phases reported in the mouse—initial (E15-P0),

intermediate (P1-P5) and final (P6-P15) stages (Lefevre *et al*, 2008). During phase 1, growth of the stereocilia rows is uniform, whereas in phase 2 differential elongation leads to the staircase pattern, with concomitant regression of supernumerary stereocilia in the mature hair bundle (Kaltenbach *et al*, 1994). Previous work has shown that clarin-1 is critically involved in formation of properly shaped IHC and OHC hair bundles (Geller *et al*, 2009; Geng *et al*, 2009, 2012; Dulon *et al*, 2018). Importantly, no such disorganization occurs in the absence of clarin-2. We show that during cochlear development and maturation, the OHC and IHC hair cell bundles in clarin-2-deficient mice appear to grow normally up to ~P5, developing their characteristic "staircase" architecture, and typical V- and U-shape organization, respectively. Indeed, by P6 almost all cochlear OHC apical circumferences have lost their immature rounded shape, to acquire a non-convex form moulded to the V-shape of the overlying hair bundle (arrowheads in Fig 4A, B, F and G). Together, these findings suggest that polarity cues (Kelly & Chen, 2007; Ezan & Montcouquiol, 2013), as well as the interactions and cellular remod-elling between supporting and sensory hair cells necessary for normal patterning of the auditory organ (Keller *et al*, 2000; Etour-nay *et al*, 2010), occur normally in the absence of clarin-2. Further-more, the persistence of immunostaining for PDZD7, a key member of the ankle link complex at the stereocilia base, correlates with the proper shaping of the V- and U-shaped hair bundles (Grati *et al*, 2012). Also, the persistence of whirlin (Fig 7B) and EPS8 (Fig EV5D), two proteins key for actin polymerization, at the tips of the differentiating stereocilia (Mburu *et al*, 2003; Manor *et al*, 2011; Zampini *et al*, 2011) are consistent with normal stereocilia elonga-tion up to at least P8 in the absence of clarin-2. Moreover, stere-ocilin is properly targeted to the distal tips of the tallest row stereocilia, indicating normal anchoring of these OHC stereocilia into the overlying tectorial membrane. Thus, we conclude that the first molecular and structural steps of hair bundle morphogenesis (initial and intermediate phases) are not affected in the absence of clarin-2.

Our qRT–PCR analyses show a postnatal increase of *Clrn2* transcripts in wild-type mouse cochleae, indicating a potential key function after the onset of hearing (~P12 in mice). We show that upon injectoporation into cochlear organs, clarin-2 localizes to stereocilia. Interestingly, only in hair cells does GFP-tagged clarin-2 target to apical surface cell membranes, indicating that addi-tional hair cell-specific co-factors are likely required for proper

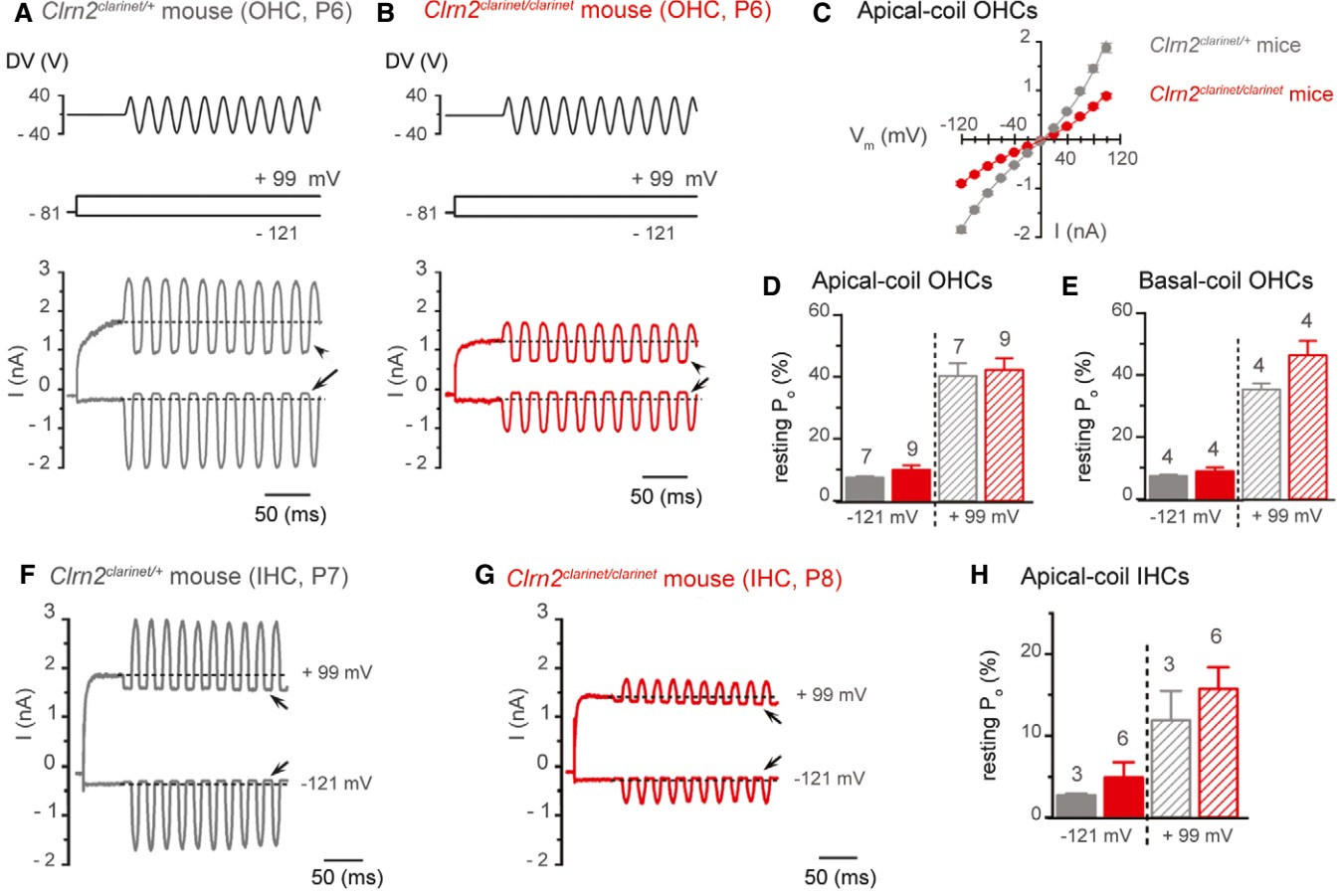

**Figure 8. Clrn2 is required for the acquisition of normal mechano-electrical transducer (MET) function in immature cochlear hair cells.**

A, B  Saturating MET currents recorded from P6 $Clrn2^{clarinet/+}$ (A) and $Clrn2^{clarinet/clarinet}$ (B) apical-coil OHCs by applying sinusoidal force stimuli of 50 Hz to the hair bundles at −121 mV and +99 mV. The driver voltage (DV) signal of ± 40 V to the fluid jet is shown above the traces (positive deflections of the DV are excitatory). The holding potential was −81 mV. Extracellular $Ca^{2+}$ concentration was 1.3 mM. Arrows and arrowheads indicate the closure of the MET currents (i.e. resting MET current) elicited during inhibitory bundle displacements at hyperpolarized and depolarized membrane potentials, respectively. Dashed lines indicate the holding current, which is the current at the holding membrane potential.

C  Average peak-to-peak current–voltage curves recorded from $Clrn2^{clarinet/+}$ (grey, P6, n = 7) and $Clrn2^{clarinet/clarinet}$ (red, P6-7, n = 9) apical-coil OHCs.

D, E  Resting open probability ($P_o$) of the MET current at the membrane potential of −121 mV and +99 mV from apical- (D) and basal-coil (E) OHCs. Number of OHCs investigated is shown above the columns. Data shown are mean ± SEM.

F, G  Saturating MET currents recorded from a P7 $Clrn2^{clarinet/+}$ (F) and a P8 $Clrn2^{clarinet/clarinet}$ (G) apical-coil IHC using the same experimental protocol described above.

H  Average $P_o$ of the MET current measured in apical-coil IHCs at the membrane potential of −121 mV and +99 mV from $Clrn2^{clarinet/+}$ (P7, n = 3) and $Clrn2^{clarinet/clarinet}$ (P7-8, n = 6) apical-coil IHCs. Data shown are mean ± SEM.

subcellular targeting of clarin-2 to the plasma membrane of stereocilia. In the $Clrn2^{clarinet/clarinet}$ mice, despite normal shape organization, the absence of clarin-2 leads to a progressive reduction in height of the middle and shortest row stereocilia, which is evident first in OHCs by P8, and then later in IHCs at P16. It is noteworthy that an abnormal shortening of mechanotransducing stereocilia has also been reported in mice deficient for several components of the mechano-electrical transduction machinery, namely the TMC1/TMC2 channel complex (Kawashima *et al*, 2011), TMIE (Zhao *et al*, 2014), LHFPL5 (Xiong *et al*, 2012), and sans or cadherin-23 (Caberlotto *et al*, 2011) and PCDH15 (Pepermans *et al*, 2014). Furthermore, Velez-Ortega and colleagues recently showed that reducing mechano-electrical transduction currents in wild-type mouse or rat hair cells, using

pharmacological channel blockers or disruption of tip links, leads to reduction in the height of the middle and shortest row "transducing" stereocilia (Velez-Ortega *et al*, 2017). Thus, it is possible that the stereocilia phenotype observed in *clarinet* mutant mice is a downstream consequence of a defect in mechano-electrical transduction. Indeed, the onset of regression of the mechanotransducing stereocilia, occurring in the absence of clarin-2, is concomitant with the loss of normal MET responses, which are already identifiable by P6-7 at both the molecular and functional levels. First, from P8 onwards, instead of the normal prolate-shaped tips of the transducing stereocilia evident in wild-type mice, which is believed to be a result of the traction force exerted by the tip link on the stereocilium apical membrane (Rzadzinska *et al*, 2004; Prost *et al*, 2007), clarin-2-deficient transducing

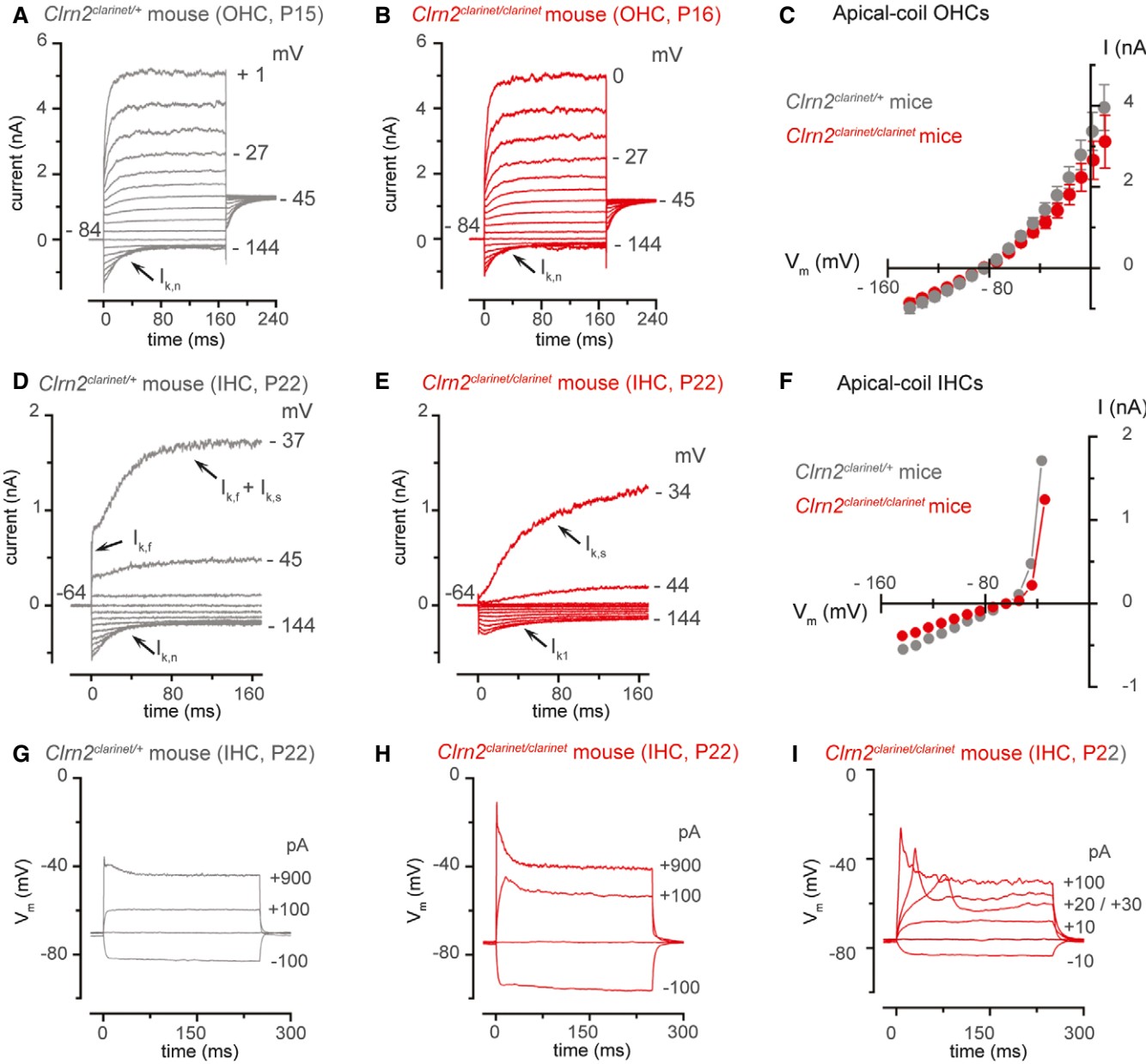

**Figure 9. IHCs, but not OHCs, fail to acquire adult-like basolateral currents in *clarinet* mice.**

A, B  Potassium currents recorded from mature *Clrn2$^{clarinet/+}$* (A, P15) and *Clrn2$^{clarinet/clarinet}$* (B, P16) apical-coil OHCs. Currents were elicited by depolarizing voltage steps (10 mV nominal increments) from −144 mV to more depolarized values from the holding potential of −84 mV. Note that the current characteristic of mature OHCs. The size of $I_{K,n}$, measured in isolation as the deactivating tail currents (difference between instantaneous and steady-state inward currents) for voltage steps from the holding potential to −124 mV, was 545 ± 115 pA (n = 5) in *Clrn2$^{clarinet/+}$* and 595 ± 67 pA (n = 7) in *Clrn2$^{clarinet/clarinet}$* OHCs.

C  Average peak current–voltage relation for the total K⁺ current recorded from *Clrn2$^{clarinet/+}$* (P15-16, n = 7) and *Clrn2$^{clarinet/clarinet}$* (P15-16, n = 5) OHCs. Data shown are mean ± SEM.

D, E  Potassium currents recorded from P22 mature *Clrn2$^{clarinet/+}$* and *Clrn2$^{clarinet/clarinet}$* apical-coil IHCs, respectively, using the same voltage protocol described above. Holding potential of −64 mV.

F  Peak current–voltage relation for the total K⁺ current recorded from the IHCs shown in panel (A) and (B).

G–I  Voltage responses from P22 *Clrn2$^{clarinet/+}$* (G) and *Clrn2$^{clarinet/clarinet}$* (H,I) IHCs. Note that Ca²⁺-dependent action potentials could be induced in mature IHCs (I).

stereocilia display round, oblate apical stereocilia wedges (Fig EV3). Second, the number of PDZD7 immunoreactive spots, essentially between the short and middle transducing stereocilia, is decreased in the absence of clarin-2, indicating loss of cytoskeleton–membrane crosslinkers that probably accompanies

the regression of the short stereocilia row (morphologically visible at later stages, about P16 in OHCs). Third, the actin-binding USH1C protein harmonin-b (Boeda *et al*, 2002), a core component of the MET machinery (Grillet *et al*, 2009; Michalski *et al*, 2009), is mislocalized in the absence of clarin-2 at P6. A similar

harmonin-b mislocation has been observed in mutant mice lacking cadherin 23- and PCDH15-composed tip links (Lefevre *et al*, 2008), which implies the necessity of mechano-electrical transduction-induced tension forces to maintain the recruitment of harmonin-b to the upper attachment point of the tip link, facing the tips of the adjacent shorter stereocilia actin core. Finally, detailed electrophysiological recordings in IHCs and OHCs at P6-8 showed that in the absence of clarin-2 the properties of the MET channel are normal, as shown by the similar resting open probability at positive membrane potentials, indicative of reduced adaptation in the unstimulated bundle from both genotypes (Corns *et al*, 2014). However, the maximal current is significantly reduced by approximately 50% in mutant IHCs and OHCs, indicating there are less available functional channels. This reduction could result from missing tip links and/or lack of sustaining tension on still present tip links, which would also be consistent with our finding of non-prolate middle and shortest row stereocilia. The presence of a functional MET current in pre-hearing IHCs, but not OHCs, has been shown to be crucial for their functional maturation into fully functional sensory receptors (Corns *et al*, 2018).

In humans, recessive *CLRN1* mutations cause Usher syndrome type IIIA (USH3A, MIM276902), characterized by post-lingual, progressive hearing loss, and variable balance and vision loss deficits (Adato *et al*, 2002). Interestingly, almost all USH3A patients develop normal speech, and a number display only mild-to-moderate hearing threshold elevation at the time of hearing loss diagnosis, at an age of 30–40 years (Ness *et al*, 2003). This contrasts with the phenotype of *Clrn1* knockout mice, where lack of clarin-1 has been shown to cause an early profound hearing loss (Geller *et al*, 2009; Geng *et al*, 2009, 2012; Dulon *et al*, 2018). Additionally, *Clrn1*$^{N48K}$ mouse mutants exhibit profound hearing loss by P25, even though USH3A patients with the *CLRN1*$^{N48K}$ mutation display post-lingual progressive hearing loss (Ness *et al*, 2003; Geng *et al*, 2012). Furthermore, *Clrn1* knockout mice do not exhibit overt retinal deficits, and so do not model this aspect of USH3A. Our characterization of the *clarinet* mutant establishes that lack of clarin-2 results in a progressive, early-onset hearing loss in mice, with no overt retinal deficits. However, given the interspecies difference in phenotypic presentation observed with *Clrn1*, we cannot exclude the possibility that pathogenic *CLRN2* mutations in humans might give rise to an Usher syndrome-like phenotype. Moreover, our studies show that clarin-2, unlike clarin-1 which is required during embryonic stages, is dispensable for the patterning and establishment of the "staircase" bundle in young postnatal hair cells. This finding supports our hypothesis that clarin-2 has an important role in functionally mature cochlear hair cells to maintain proper integrity of the transducing stereocilia. Accordingly, we investigated the UK Biobank cohort and found that genetic variation at the human *CLRN2* locus is highly associated with adult hearing difficulty in this cohort.

Our analysis revealed a cluster of SNPs that lie within, or close to, the *CLRN2* gene, which are significantly associated with an adult hearing difficulty phenotype. Within the 20 most highly associated SNPs, the majority are either intronic or intergenic (Table EV1). As such, it is probable that these do not directly affect *CLRN2* expression. Instead, these SNPs are likely in linkage disequilibrium with an, as yet unidentified, causal variant(s) within the UK Biobank population cohort. However, the second

most associated SNP (rs13147559) is located within the *CLRN2* gene coding sequence (c.337C > G), with presence of the minor allele causing a leucine-to-valine missense variation at codon 113 (p.Leu113Val). This leucine residue, based on comparison to the 3D modelling prediction of hsCLRN1 (Gyorgy *et al*, 2019), is located within the second transmembrane domain of hsCLRN2 and is evolutionarily conserved across species. While this variation involves two hydrophobic amino acids that possess similar structures, valine does have a shorter side chain. Furthermore, prediction tools suggest that this substitution might be detrimental to protein function returning scores of "possibly damaging" and "medium". Interestingly, similar substitutions located in the highly conserved transmembrane domains of presenilin, encoded by the gene *PSEN1*, have been reported in patients with Alzheimer's disease. These missense variants (p.Leu250Val and p.Leu153Val) have been proposed to interfere with the helix alignment of the transmembrane domain altering protein optimal activity, thus accounting for disease expression (Furuya *et al*, 2003; Larner, 2013). However, additional studies are needed to determine whether the hsCLRN2 p.Leu113Val missense variant is causal of, or merely associated with, the adult hearing difficulty trait. Perhaps it may be that "mild" *CLRN2* hypomorphic mutations, such as p.Leu113Val may represent, is likely to predispose to a progressive, late-onset hearing loss phenotype. Conversely, it is possible that more pathogenic *CLRN2* mutations may elicit a more severe, early-onset hearing loss phenotype. There are examples of this, for instance *TMPRSS3*, encoding transmembrane protease serine 3, has been reported to cause severe-to-profound prelingual hearing loss (DFNB10) as well as progressive hearing impairment with post-lingual onset (DFNB8) due to differential pathogenic mutations (Gao *et al*, 2017). Of note, a recent work by Gopal S. and colleagues reported a recessively inherited non-syndromic sensorineural hearing loss in a consanguineous Iranian family caused by a *CLRN2* mutation that results in a missense mutation in the encoded protein (Gopal *et al*, 2019). Affected patients develop post-lingual moderate-to-profound hearing loss with no indication of balance or vision deficits. Altogether, while mutations in *CLRN1* unambiguously lead to USH3A, current findings suggest that *CLRN2* mutation most likely causes non-syndromic hearing loss. However, additional cases need to be identified to clarify the genotype–phenotype relationship between the impaired extent of activity of the mutated clarin-2 protein, the age of onset, the severity and the extent of the disease phenotype. Nonetheless, our study demonstrates the utility of interrogating human large cohort study data as a means to help validate candidate genes arising from forward genetic, or whole-genome sequencing, screens.

## Conclusion

*Clrn2*$^{clarinet}$ mice are ENU-induced mutants that exhibit early-onset sensorineural hearing loss, associated with a nonsense mutation in the encoded tetraspan protein clarin-2. Utilizing these mice, we demonstrate that clarin-2 is required for the maintenance of stereocilia bundle morphology, and show that its loss leads to decreased mechano-electrical transduction and progressive hearing impairment. Moreover, utilizing data from the UK Biobank study, *CLRN2* is identified as a novel candidate gene for human non-syndromic progressive age-related hearing loss. Our study of the *clarinet* mouse

mutant provides insight into the interplay between mechano-electrical transduction and stereocilia bundle maintenance.

# Materials and Methods

### Mice

The *clarinet* mutant was identified from the MRC Harwell Institute phenotype-driven *N*-ethyl-*N*-nitrosourea (ENU) Ageing Screen (Potter *et al*, 2016). In this screen, ENU-mutagenized C57BL/6J males were mated with wild-type "sighted C3H" (C3H.Pde6b+) females (Hoelter *et al*, 2008). The resulting $G_1$ males were crossed with C3H.Pde6b+ females to produce $G_2$ females, all of which were screened for the $Cdh23^{ahl}$ allele (Johnson *et al*, 2006). $Cdh23^{+/+}$ $G_2$ females were then backcrossed to their $G_1$ fathers to generate recessive $G_3$ pedigrees, which entered a longitudinal phenotyping pipeline. Auditory phenotyping comprised clickbox testing at 3, 6, 9 and 12 months of age and ABR at 9 months of age. Initial linkage was determined using SNP mapping (Tepnel Life Sciences), delineating a critical interval containing the *clarinet* mutation on Chromosome 5. Whole-genome sequencing was undertaken using DNA from an affected $G_3$ mouse (Oxford Genomics Centre, Wellcome Trust Centre for Human Genetics, University of Oxford) and putative lesions validated by standard PCR and Sanger sequencing. $Clrn2^{clarinet/+}$ carrier mice were subsequently backcrossed to C57BL/6J for ten generations until congenic. The $Clrn2^{del629}$ mutant line was generated on a C57BL/6N background by the Molecular and Cellular Biology group at the Mary Lyon Centre (MLC), MRC Harwell Institute using CRISPR-Cas9 genome editing (Mianne *et al*, 2016; Codner *et al*, 2018) (Table EV2). Within the MLC, all mice were housed and maintained under specific pathogen-free conditions in individually ventilated cages, with environmental conditions as outlined in the Home Office Code of Practice. Animals were housed with littermates until weaned and then housed with mice of the same gender and of similar ages, which was often their littermates. Both male and female animals were used for all experiments.

Animal procedures at the MRC Harwell Institute and University of Sheffield were licensed by the Home Office under the Animals (Scientific Procedures) Act 1986, UK, and additionally approved by the relevant Institutional Ethical Review Committees. Animal procedures at the Institut Pasteur were accredited by the French Ministry of Agriculture to allow experiments on live mice [accreditation 75-15-01, issued on 6 September 2013 in appliance of the French and European regulations on care and protection of the Laboratory Animals (EC Directive 2010/63, French Law 2013-118, 6 February 2013)]. Protocols were approved by the veterinary staff of the Institut Pasteur animal facility and were performed in compliance with the NIH Animal Welfare Insurance #A5476-01 issued on July 31, 2012.

### Association analysis of human hearing with *CLRN2* variation in the UK Biobank Cohort

The cohort used for the human association analysis consisted of 500,000 UK Biobank (UKBB) participant (Sudlow *et al*, 2015) with "White British" ancestry. Samples with excess heterozygosity, excess relatedness and sex discrepancies were identified and

removed prior to analysis. Samples were genotyped on one of two arrays; 50,000 samples were genotyped on the Affymetrix UK BiLEVE Axiom array, while the remaining 450,000 were genotyped on the Affymetrix UK Biobank Axiom® array. The two arrays shared 95% coverage resulting in > 800,000 genotyped SNPs. Imputation was carried out centrally by UKBB, primarily using the HRC reference panel and IMPUTE2 (Howie *et al*, 2009, 2011). Analysis in this study was conducted with version 3 of the UK Biobank imputed data.

For association testing, cases and controls were defined based on participants' responses to questions regarding hearing ability. Briefly, participants who answered YES to both "Do you have any difficulty with your hearing?" and "Do you find it difficult to follow a conversation if there is background noise (such as TV, radio, children playing)?" were defined as cases, $n = 102,832$. Those who answered NO to the same questions were defined as controls. Any individuals who said they used a hearing aid were removed from the control group. Finally, individuals below the age of 50 years of age were removed from the controls to ensure a similar age distribution to cases, resulting in a sample with $n = 163,333$ for the control group.

A linear mixed models approach was used to test for association for all SNPs within 100 kb of the *CLRN2* gene using BOLT-LMM v.2[3] (Loh *et al*, 2018) for the association analysis, which corrects for population stratification and within-sample relatedness. In addition, analysis was adjusted for age, sex, UK Biobank genotyping platform and UK Biobank PCs1-10. For quality control, SNPs were filtered based on the two thresholds: (i) minor allele frequency > 0.01 (ii) INFO score > 0.7. Individuals with < 98% genotype call rate were removed. Following genomic quality control filters and selection for White British samples, association analysis was performed on the remaining 87,056 cases and 163,333 controls. To adjust for multiple testing, the Bonferroni-adjusted significance threshold for this analysis is set at 0.0009 based on calculating the effective number of independent SNPs within this region ($n = 55$) (Li *et al*, 2012).

Utilization of the UK Biobank Resource was conducted under Application Number 11516.

### Auditory phenotyping

Auditory brainstem response (ABR) tests were performed using a click stimulus and frequency-specific tone-burst stimuli (at 8, 16 and 32 kHz) to screen mice for auditory phenotypes and investigate auditory function (Hardisty-Hughes *et al*, 2010). Mice were anaesthetized by intraperitoneal injection of ketamine (100 mg/ml at 10% v/v) and xylazine (20 mg/ml at 5% v/v) administered at the rate of 0.1 ml/10 g body mass. Once fully anaesthetized, mice were placed on a heated mat inside a sound-attenuated chamber (ETS Lindgren) and recording electrodes (Grass Telefactor F-E2-12) were placed subdermally over the vertex (active), right mastoid (reference) and left mastoid (ground). ABR responses were collected, amplified and averaged using TDT System 3 hardware and BioSig software (Tucker Davies Technology, Alachua, FL, USA). The click stimulus consisted of a 0.1-ms broadband click presented at a rate of 21.1/s. Tone-burst stimuli were of 7-ms duration including rise/fall gating using a 1-ms Cos2 filter, presented at a rate of 42.5/s. All stimuli were presented free-field to the right ear of the mouse,

starting at 90 dB SPL and decreasing in 5 dB increments until a threshold was determined visually by the absence of replicable response peaks. For graphical representation, mice not showing an ABR response at the maximum level tested (90 dB SPL) were recorded as having a threshold of 95 dB SPL. These mice/thresholds were included when calculating genotype average thresholds. All ABRs were performed blind to genotype, to ensure thresholds were obtained in an unbiased manner. Mice were recovered using 0.1 ml of anaesthetic reversal agent atipamezole (Antisedan™, 5 mg/ml at 1% v/v), unless aged P16, when the procedure was performed terminally.

Distortion product otoacoustic emission tests were performed using frequency-specific tone-burst stimuli from 8 to 32 kHz with the TDT RZ6 System 3 hardware and BioSig RZ software (Tucker Davis Technology, Alachua, FL, USA) software. An ER10B+ low-noise probe microphone (Etymotic Research) was used to measure the DPOAE near the tympanic membrane. Tone stimuli were presented via separate MF1 (Tucker Davis Technology) speakers, with f1 and f2 at a ratio of f2/f1 = 1.2 (L1 = 65 dB SPL, L2 = 55 dB SPL). Mice were anaesthetized via intraperitoneal injection of ketamine (100 mg/ml at 10% v/v), xylazine (20 mg/ml at 5% v/v) and acepromazine (2 mg/ml at 8% v/v) administered at a rate of 0.1 ml/10 g body mass. Once surgical anaesthesia was confirmed by the absence of a pedal reflex, a section of the pinna was removed to enable unobstructed access to the external auditory meatus. Mice were then placed on a heated mat inside a sound-attenuated chamber (ETS Lindgren), and a pipette tip containing the DPOAE probe assembly was inserted into the ear canal. In-ear calibration was performed before each test. The f1 and f2 tones were presented continuously, and a fast-Fourier transform was performed on the averaged response of 356 epochs (each ~21 ms). The level of the 2f1-f2 DPOAE response was recorded and the noise floor calculated by averaging the four frequency bins either side of the 2f1-f2 frequency.

### *Clrn2* expression in tissues and isolated auditory hair cells

To allow analysis of the $Clrn2^{del629}$ allele, whole cochlear ducts were collected from P21 (+/− 1 day) $Clrn2^{+/+}$, $Clrn2^{+/del629}$ and $Clrn2^{del629/del629}$ littermate mice, and stored in RNA *later* Stabilising Solution (Invitrogen) at −20°C until processed. Total RNA was extracted using TRIzol Reagent (Invitrogen) and used as template for cDNA generation using the High Capacity cDNA Reverse Transcription kit (Applied Biosystems). Subsequent PCR amplification was undertaken utilizing *Clrn2*-specific primers: *Clrn2-Exon 1 For* (CTCATTAGTATGCCTGGATGG)/*Clrn2-Exon 3 Rev* (TTAGTCTT GATTTCTGGAAGGG) and then electrophoresed on a 2% agarose gel. PCR products were excised and purified using the GENECLEAN II kit (MP Biomedical) and subject to Sanger sequencing (Oxford Source Bioscience). Data were analysed using the SeqMan Pro (DNASTAR) software. For RT–PCR analyses of tissues and hair cells, fresh tissues (inner ear, eye, brain and muscle) of P30 wild-type C57BL/6J mice and isolated auditory hair cells from P15 mice were collected and quickly frozen in liquid nitrogen and stored at −80°C until processing. Auditory hair cells were isolated under direct visual microscope observation. Only solitary IHCs and OHCs identified based on their typical morphology (cylindrical OHCs and pear-shaped IHCs) were taken into consideration, hair cells with ambiguous morphology were excluded. Total RNAs were isolated

with TRIzol Reagent (Invitrogen) according to the manufacturer's instructions. Total RNA (400 ng) was reverse-transcribed with the Superscript One-Step RT–PCR system (Invitrogen). For tissue-specific *Clrn2* expression studies, the primers employed were as follows: *Clrn2-F1* (*ATGCCTGGATGGTTCAAAAAG*)/*Clrn2-R1* (*TCAC AAGGTGTACGCAGGAGTCAG*), and a β-actin control: *β-actin-F* (*ACC TGACAGACTACCTCAT*)/*β-actin-R* (*AGACAGCACTGTGTTGGCAT*). For hair cell type-specific *Clrn2* expression, the primers employed were as follows: *Clrn2-F2* (*GGGACGCCAGTCCCAATTTA*)/*Clrn2-R2* (*ACTCCACCTGCGAGGACATT*), with hair cell-specific controls: *Otoferlin* (IHC positive control) *Otof-F* (*CATCGAGTGTGCAGG AAAGG*)/*Otof-R* (*ACCTGACCACAGCATCAGA*); and *Oncomodulin* (OHC positive control) *Ocm-F* (*CGGCCCTGCAGGAATGCCAA*)/*Ocm-R* (*GCTTCAGGGGGACTTGGTAAA*). PCR products were separated by electrophoresis on 2% agarose gels.

### Behavioural tests

Multiple behavioural tests were used to assess the vestibular function of *clarinet* mice, as described previously (Hardisty-Hughes *et al*, 2010). In the platform test, mice were placed on a small platform (7 × 7 cm, at a height of 29 cm) and the time on the platform was recorded over a period of 2 min. The contact righting test consisted of placing a mouse in a closed transparent tube and determining whether it was able to successfully regain standing position upon a 180° rotation of the tube (score 1) or fail (score 0). For the swim test, each mouse was placed in a container filled with water at 22–23°C and given a score determined as follows: score 0 = normal swimming; score 1 = irregular swimming; score 2 = immobile floating; score 3 = underwater tumbling.

### Electroretinogram response measurements

To measure electroretinograms (ERGs), animals were kept in the dark to adapt to darkness overnight as previously described (Michel *et al*, 2017). Each mouse was anesthetized with a mixture of ketamine (80 mg/kg, Axience, France) and xylazine (8 mg/kg, Axience, France), and placed over a warming pad to maintain body temperature at 37°C. Their pupils were dilated with tropicamide (Mydriaticum; Théa, Clermont-Ferrand, France) and phenylephrine (Neosynephrine; Europhta, Monaco). The cornea was locally anesthetized with oxybuprocaine chlorhydrate (Théa, Clermont-Ferrand, France). Upper and lower lids were retracted to keep eyes open and bulging. Retinal responses were recorded with a gold-loop electrode brought into contact with the cornea through a layer of lubrithal (Dechra, France), with needle electrodes placed in the cheeks and back used as reference and ground electrodes, respectively (Yang *et al*, 2009). The light stimuli were provided by an LED in a Ganzfeld stimulator (SIEM Bio-médicale, France). Responses were amplified and filtered (1 Hz-low and 300 Hz-high cut-off filters) with a one-channel DC-/AC amplifier. One level of stimulus intensity (8 cd.s/m$^2$) was used for scotopic ERG recording. Each of the response obtained was averaged over five flash stimulations. Photopic cone ERGs were recorded in a rod-suppressing background light of 20 cd.s/m$^2$, after a 5-min adaptation period. An 8 cds/m$^2$ level of stimulus intensity was used for the light-adapted ERGs. Each cone photopic ERG response presented was averaged over ten consecutive flashes.

### Immunolabelling

For the synaptic labelling experiments, mice were culled by cervical dislocation and inner ears were fixed in 4% paraformaldehyde (PFA) in PBS for 1 h at room temperature (RT). Post-fixation, ears were fine dissected to expose the sensory epithelium then permeabilized using 0.1% Triton X-100 in PBS for 10 min at RT. Samples were blocked in 5% donkey serum (Sigma) for 1 h at RT and immunolabelled with primary antibodies overnight at 37°C. To enable detection, samples were incubated with fluorophore-coupled secondary antibodies for 1 h at 37°C then stained with DAPI (1:2,500, Thermo Fisher) for 5 min at RT. Samples were mounted onto slides in SlowFade® Gold (Life Technologies) and visualized using a Zeiss LSM 710 fluorescence confocal microscope. Primary antibodies: rabbit anti-Ribeye (Synaptic Systems; 192103; 1:200); mouse anti-GluR2 (Millipore; MABN1189; 1:200). Secondary antibodies: Alexa Fluor® donkey anti-rabbit 568 (Invitrogen; 1:200); Alexa Fluor® donkey anti-mouse 488 (Invitrogen; 1:500).

For all other immunofluorescence experiments, samples were processed as previously described (Michel et al, 2017). Briefly, for cochlear whole-mount preparations, micro-dissected, fixed mouse organs of Corti (4% PFA in PBS, pH 7.4 for 1 h at RT) were rinsed, then blocked by incubation in PBS supplemented with 20% normal goat serum and 0.3% Triton X-100 for 1 h at RT. After incubation with primary antibodies in PBS 1% bovine serum albumin overnight at 4°C, samples were rinsed in PBS then incubated with specific secondary antibodies (and phalloidin for actin staining when required) for 1 h at RT. They were then immersed in DAPI (Sigma) for nuclear labelling and rinsed before mounting using Fluorsave (Calbiochem, La Jolla, CA).

To test the occurrence of apoptosis in the retina, cryosections from control and clarinet mice were analysed using the in-situ Cell Death Detection Kit, Fluorescein (Roche), according to the manufacturer's instructions.

To detect clarin-2 protein, the cDNA encoding amino acid residues 159–232 (VKFHDLTERIANFQERLFQFVVVEEQYEESFWIC VASASAHAANLVVVAISQIPLPEIKTKMEEATVTPEDILY) of mouse Clrn2 were amplified by PCR, cloned into pRSET (Life Technologies) and transformed into BL21(DE3)pLysS competent cells (Life Technologies) for protein production. The fusion protein was purified using cobalt chloride-charged chelating sepharose fast flow resin (GE Healthcare Life Sciences) and used to immunize rabbits (Covalab). The antisera were purified by affinity chromatography using the fusion protein (antigen) coupled to SulfoLink resin (Pierce), according to the manufacturer's instructions. We checked the specificity of the affinity-purified antibodies by immunofluorescence using transfected cells and mouse organs of Corti (clarinet samples were used as negative controls). The purified homemade, but not commercial, anti-clarin-2 antibodies did detect both GFP- and FLAG-tagged clarin-2 in transfected cells (Fig EV4B–D). However, repeated attempts to detect endogenous clarin-2 in the mouse auditory sensory organ at different postnatal stages, under various conditions of fixation and antigen-retrieval, were unsuccessful. Similar observations were made using commercially available rabbit polyclonal anti-clarin-2 antibodies: the anti-clarin-2 from Proteintech (1:100, 23994-1-AP) and from Atlas (HPA042407; 1:100). The absence of specific immuno-detection in hair cells could reflect a low level of endogenous expression (Fig EV4).

To detect PDZD7, we used a newly generated homemade polyclonal rabbit antibody. It is derived against a mouse PDZD7 fusion protein (amino acid 2–83, accession number NP_001182194.1) and has been validated in transfected cells and mouse organs of Corti. The following other primary antibodies were used: rabbit anti-myosin VIIa (1:500), rabbit anti-harmonin-b (1:50) and rabbit anti-whirlin (1:100) (Sahly et al, 2012). Other primary antibodies were used: rabbit anti-stereocilin (1:150)(Verpy et al, 2008), mouse anti-EPS8 (1:200; 610144; BD Bioscience) mouse and rabbit anti-CtBP2 (1:200; Goat polyclonal, Santa Cruz, USA; SC-5966) to detect the ribbon protein ribeye, rabbit anti-GluR2/3 (1:200; Millipore), rabbit anti-opsin, blue (1:100; AB5407, Merck-Millipore), mouse anti-Iba1 (1:200; MABN92, Merck-Millipore), mouse anti-rhodopsin (1:500; MAB5316, Merck-Millipore), mouse anti-FLAG2 (1:120, F3165, Sigma-Aldrich) and rabbit anti-GFP (1:250, Invitrogen). The following specific secondary antibodies were used: ATTO 488-conjugated goat anti-rabbit IgG (1:500, Sigma-Aldrich) and ATTO 500 goat anti-mouse IgG antibody (1:500, Sigma-Aldrich). ATTO 565 phalloidin (1:700; Sigma-Aldrich) was also used to label F-actin.

Samples were imaged at RT with a confocal microscope (LSM 700; Carl Zeiss) fitted with a Plan-Apochromat 63× NA 1.4 oil immersion objective from Carl Zeiss. To quantify the positioning of the harmonin-b immunoreactive area relative to the tip of the taller stereocilium, we used ImageJ software (NIH). Quantification was made using IHC, rather than OHC, bundles as it is easier to define individual stereocilia, allowing accurate measurements of the distance of the harmonin-b immunoreactive spot relative to the tip of the stereocilium. For each clarinet (n = 4) and wild-type (n = 4) mouse, confocal images of 8–10 IHCs with well-preserved tallest stereocilia were considered for measurements. For each cell, the distance value taken into consideration is an average value from 3 stereocilia of the same cell: for each stereocilium, we measure the distance between the stereocilium tip to the centre of harmonin-b-immunoreactive spot located on the side of the same stereocilium. Measurements were analysed using Student's t-test.

### Scanning electron microscopy

Mice were euthanized by cervical dislocation, and inner ears were removed and fixed in 2.5% glutaraldehyde (TAAB Laboratories Equipment Ltd.) in 0.1 M phosphate buffer (Sigma-Aldrich) overnight at 4°C. Following decalcification in 4.3% EDTA, cochleae were sub-dissected to expose the sensory epithelium then "OTO processed" with alternating incubations in 1% osmium tetroxide (TAAB Laboratories Equipment Ltd.) in 0.1 M sodium cacodylate (Sigma-Aldrich) and 1% thiocarbohydrazide (Sigma-Aldrich) in ddH$_2$O. Ears were dehydrated through a graded ethanol (Fisher Scientific) series (25–100%) at 4°C and stored in 100% acetone (VWR Chemicals) until critical point drying with liquid CO$_2$ using an Emitech K850 (EM Technologies Ltd). Ears were mounted onto stubs using silver paint (Agar Scientific), sputter coated with palladium using a Quorum Q150R S sputter coater (Quorum Technologies) and visualized with a JSM-6010LV Scanning Electron Microscope (JEOL).

To analyse prolateness of the stereocilia tips between control and clarin-2-deficient hair bundles, scanning electron micrographs were utilized and small square selections framing the tips of individual

second row stereocilia were delimited and extracted to create an array of tip images using ImageJ (Fiji). Consistency between the images captured was maintained by ensuring the side of the square selection was parallel to the distal edge of the stereocilium, and ensuring that a constant distance was kept from the upper edge. Stereocilia from OHC bundles were imaged at a ~45° angle from the perspective plane. All images were taken to scale after calibration to the scanning electron microscope software-generated scale bar. The image array for each genotype was then loaded as a pseudo z-stack and a median-intensity z-projection generated. For each projection, at least 80 tip images per genotype were used—3 animals per genotype, 3 bundles per animal, with all bundles from the mid-region of the cochlea. These projections were processed using auto-tone and then pseudo-coloured in Adobe Photoshop, before a perimeter outline was drawn using Adobe Illustrator in order to show the shape of the median Z-projected stereocilia tip (see Fig EV3).

To assess stereocilia heights, at least three ears (one ear per mouse) were analysed for each genotype at each time point. Duplicate images of the middle turn (180° to 360°) of the cochlea were taken, with a 5° tilt between them. IHC bundles were imaged at 8,000× magnification and OHC bundles at 15,000× magnification, all at a constant working distance of 20 μM. Three different bundles were analysed per animal, with up to nine different measurements taken per bundle: three measurements from the tallest row of stereocilia, three from the middle row and three from the shortest row (if present). IHC and OHC measurements were obtained using ImageJ software (NIH) and corrected using a pseudo-eucentric tilting approach (Bariani *et al*, 2005). A single measure $x_1$ (e.g. length of the tallest row of one stereocilium) was taken on a first micrograph and measured again ($x_2$) on the corresponding 5°-tilted repeat micrograph. Perpendicular countermeasures ($y_1$ and $y_2$) to $x_1$ and $x_2$ were also taken in every instance. Later, countermeasures were fed onto equation (1), in order to estimate uncertainty ($\zeta$) due to plane rotation. Every pair of tilted-coupled measures ($x_1$ and $x_2$) was then fed onto equation (2), along with the uncertainty estimate ($\zeta$) from equation (1), thus obtaining a close approximation ($\xi$) of the true measure of the structures investigated.

$$\zeta = \frac{(\Delta y)\cos\Delta\varphi + (2y_1(y_1 - \Delta y)/d)\sin\Delta\varphi}{(1 + y_1(y_1 - \Delta y)/d^2\sin\Delta\varphi) + (\Delta y/d)\cos(2\Delta\varphi)}; \qquad (1)$$

$$\xi = \frac{2d - 2\zeta\cos\Delta\varphi}{d/x_1 + d/x_2}; \qquad (2)$$

where, $\zeta$ = uncertainty estimate; $y_{1,2}$ = perpendicular countermeasures to measures $x_{1,2}$; $\Delta y$ = arithmetic difference of countermeasures $y_1$ and $y_2$; $\Delta\varphi$ = tilting angle (5°); $d$ = working distance (20 μM); $x_{1,2}$ = tilted paired-measures of structure of interest; $\xi$ = estimate of true size of structure of interest.

### *Clrn2* expression in whole cochlea

Total RNA samples were extracted from wild-type C57BL/6J mouse whole cochleae at E17.5 (embryonic day 17.5), P4, P8, P12, P16 and P28 using a Direct-zol RNA MiniPrep Kit (Zymo Research). Five separate samples were prepared for each age, comprising whole-cochlea mouse RNA extracted from both ears. In each case, quintuplicate samples from independent mice were analysed for each stage. Complementary DNAs were generated from 1 μg total RNA using the High Capacity cDNA synthesis kit (Applied Biosystems) following the manufacturer's instructions. Amplifications were performed using TaqMan Gene Expression Assays with Fast Universal PCR Master Mix. A β-actin primer/probe set (*Mm00607939_s1*) was used as an internal control, and a custom TaqMan assay was designed to allow the specific amplification of the *Clrn2* transcripts using the following forward and reverse primers, and TaqMan Gene Expression Assay: *Clrn2*-F 5′-*AAGATGTC CACTTGCCCAAC*-3′, *Clrn2*-R 5′-*GACCAGGGTTCTTGTGCTTC*-3′, *Mm03990594_m1*. PCRs were run in triplicate in the Applied Biosystems real-time PCR device (7500 Fast Real-Time PCR System) in 20-μl reactions containing 10 μl Fast TaqMan Master Mix, 1 μl TaqMan assay, 4 μl ddH$_2$O and 5 μl complementary DNA (5 ng/μl), using the following cycles: 95°C for 20 s and 40 cycles at 95°C for 3 s and 60°C for 30 s. Amplification data were recorded and relative quantification performed using the 7500 software version 2.0.6 software (Applied Biosystems) comparing threshold cycles (C$_t$). *Clrn2* mRNA levels were first normalized to *Actin* ($\Delta$Ct = Ct$_{Nptn}$ − Ct$_{Actin}$) at each age, and changes in expression relative to P4 were calculated as $2^{-(\Delta Ct - \Delta CtP4)}$.

### Injectoporation

The injectoporation experiments of cochlear explants were performed as previously described (Xiong *et al*, 2014), with a few modifications. In brief, cochleae were dissected from P2 wild-type mice, cut into 4–6 pieces and cultured for 6 h in DMEM/F12 medium with 100 ng/μl ampicillin. Next, adherent cochlear explants were placed between two platinum wire electrodes (Sure-pure Chemetals) and a patch pipette (2 μm diameter) placed between the second and third row of OHCs was used to deliver the plasmid (1 μg/μl in 1× HBSS) to the hearing organ. The pipette and electrode were positioned using an Axioscope 2 Carl Zeiss microscope with a 40× objective (Olympus) and two micromanipulators (Sutter MPC-200). To trigger plasmid entry into cells, 3 square-pulses with a magnitude of 60 V (15 ms length, 1-s intervals) were applied, using an ECM 830 electroporator (Harvard Apparatus). Organs of Corti were cultured for another 12 h in DMEM/F12 and fixed in 4% PFA for 1 h before immunostaining. Samples were analysed at RT with a confocal microscope (LSM 700; Carl Zeiss) fitted with a Plan-Apochromat 63× NA 1.4 oil immersion objective from Carl Zeiss.

### Electrophysiology

Electrophysiological recordings were made from apical-coil OHCs and IHCs of *clarinet* mice aged P6-22. A number of basal-coil OHCs were also investigated at P7-8. Cochleae were dissected in normal extracellular solution (in mM): 135 NaCl, 5.8 KCl, 1.3 CaCl$_2$, 0.9 MgCl$_2$, 0.7 NaH$_2$PO$_4$, 5.6 ᴅ-glucose, 10 HEPES-NaOH. Sodium pyruvate (2 mM), MEM amino acids solution (50X, without ʟ-Glutamine) and MEM vitamins solution (100×) were added from concentrates (Fisher Scientific, UK). The pH was adjusted to 7.5 (308 mosmol/kg). The dissected cochleae were transferred to a microscope

chamber, immobilized with a nylon mesh (Corns *et al*, 2016) and continuously perfused with a peristaltic pump using the above extracellular solution. The organs of Corti were viewed using an upright microscope (Leica DMLMF, Germany; Nikon FN1, Japan) with Nomarski optics (x60 or x63 objectives).

MET currents were elicited by stimulating the hair bundles of OHCs and IHCs using a fluid jet from a pipette (tip diameter 8–10 μm) driven by a piezoelectric disc (Corns *et al*, 2014). The pipette tip of the fluid jet was positioned near to the bundles to elicit a maximal MET current. Mechanical stimuli were applied as 50 Hz sinusoids (filtered at 0.25 kHz, 8-pole Bessel) with driving voltages of $\pm$ 40 V. MET currents were recorded with a patch pipette solution containing (in mM): 106 Cs-glutamate, 20 CsCl, 3 $MgCl_2$, 1 EGTA-CsOH, 5 $Na_2ATP$, 0.3 $Na_2GTP$, 5 HEPES-CsOH, 10 sodium phosphocreatine (pH 7.3). Membrane potentials were corrected for the liquid junction potential (LJP) of $-11$ mV, measured between electrode and bath solutions.

Patch clamp recordings were performed using an Optopatch (Cairn Research Ltd, UK) amplifier. Patch pipettes were made from soda glass capillaries with a typical resistance in the extracellular solution of 2–3 MΩ. In order to reduce the electrode capacitance, patch electrodes were coated with surf wax (Mr Zoggs SexWax, USA). Basolateral membrane recordings were performed using an intracellular solution containing (in mM): 131 KCl, 3 $MgCl_2$, 1 EGTA-KOH, 5 $Na_2ATP$, 5 HEPES-KOH, 10 $Na_2$-phosphocreatine (pH 7.3; osmolality ~296 mmol/kg). Data acquisition was controlled by pClamp software using Digidata 1440A boards (Molecular Devices, USA). Recordings were low-pass filtered at 2.5 kHz (8-pole Bessel), sampled at 5 kHz and stored on computer for off-line analysis (Origin: OriginLab, USA). Membrane potentials in voltage clamp were corrected for the voltage drop across the uncompensated residual series resistance and for a LJP of $-4$ mV measured between electrode and bath solutions. All recordings were performed at RT.

### Statistical analysis

Unless stated, data were analysed using Student's two-tailed *t*-test (two study groups) or one-way ANOVA followed by Tukey's post-hoc test (three study groups) using GraphPad Prism. Graphs are presented as mean $\pm$ SD, unless stated, and $P < 0.05$ indicates statistical significance. Using data obtained from the original MPC169 cohort, we used GraphPad StatMate to carry out power calculations to determine sample size in an unpaired *t*-test using the standard deviation of the measured hearing thresholds, a significance level of $P = 0.01$ (two-tailed) and a power of 95%. As such, an effect size was estimated using real data corresponding to the *clarinet* hearing loss phenotype comparing hearing thresholds of wild-type and homozygous littermates. This determined that a cohort size of $\geq 4$ mice/genotype would be sufficient to detect an auditory threshold difference. All phenotyping was performed blind to genotype, and no data were excluded from analysis.

### Availability of materials

$Clrn2^{clarinet}$ and $Clrn2^{del629}$ mice are available on request from the MRC Harwell Institute.

### The paper explained

#### Problem

Hearing loss is a very prevalent condition that can result from environmental causes, genetic predisposition or an interaction of both. Over the last two decades, while much progress has been made in understanding the genetic bases of congenital and early-onset hearing loss, we have only just begun to elaborate upon the genetic landscape of age-related hearing loss. Going forward, increased knowledge of the genes and molecular pathways required for the maintenance of hearing function will likely provide opportunities for the design of therapeutic approaches to prevent progressive hearing loss.

#### Results

Utilizing a forward genetic screen in mice, we identified a mutation within the *Clrn2* gene as the cause of hearing loss in the *clarinet* mutant. Loss of the encoded tetraspan-like protein clarin-2 leads to an early-onset, progressive hearing loss. Interestingly, analysis of a large cohort of patients with adult hearing difficulties from the UK Biobank study we identify *CLRN2* as a novel candidate gene for human non-syndromic progressive age-related hearing loss. Our morphological, molecular and functional investigations of the clarin-2-deficient mice establish that while the protein is not required for the initial formation of cochlear sensory hair cell stereocilia bundles, it is critical for maintaining normal bundle integrity and functioning. In the differentiating hair bundles, lack of clarin-2 leads to loss of mechano-electrical transduction, followed by selective progressive loss of the transducing stereocilia.

#### Impact

Integrated mouse and human approaches continue to elaborate upon our understanding of the genetic mechanisms required for mammalian hearing. In this study, we establish that *Clrn2* is a novel deafness gene associated with progressive hearing loss in both mice and humans, and as such, severe loss-of-function *CLRN2* mutations should be considered in the aetiology of human autosomal recessive hearing loss. Moreover, our findings demonstrate a key role for clarin-2 in mammalian hearing, providing insights into the interplay between mechano-electrical transduction and stereocilia architecture maintenance.

**Expanded View** for this article is available online.

### Acknowledgements

We would like to thank L. Vizor, R. Kent, M. Hutchison, S. Nouaille and MLC Ward staff for their assistance with the breeding of *clarinet* and *del629* animals; Y. Bouleau for help with isolation of hair cells; J. Sanderson for his technical assistance with the immunolabelling experiments; and the Molecular and Cellular Biology group at the Harwell Institute for generating the $Clrn2^{del629}$ mice. This work was supported by: Medical Research Council (MC_U142684175 to S.D.M.B. and MC_UP_1503/2 to M.R.B), Wellcome Trust (102892 to W.M.); the French National Research Agency (ANR) as part of the second "Investissements d'Avenir" programme (light4deaf, ANR-15-RHUS-0001) and LHW-Stiftung (to C.P. & A.E.); ANR-HearInNoise-(ANR-17-CE16-0017 to A.E.); NIDCD/NIH (R01DC013817), NIMH/NIH (R24MH114815) and the Hearing Restoration Program of the Hearing Health Foundation (to R.H.); and an Action on Hearing Loss PhD studentship (to F.W. and S.Da.), which was supported by the National Institute for Health Research University College London Hospitals Biomedical Research Centre. L.D. is a Medical Research Council PhD student. P.P. benefited from a fellowship from the European Union's Horizon 2020

Marie Sklodowska-Curie grant No 665807. S.L.J. is a Royal Society University Research Fellow.

## Author contributions

LAD, PP, CA, PM, AP, DW, CTE, MMS, SDe, RH, DD, WM, AE and MRB designed and interpreted the experiments. LAD, PP, CA, PM, SDe, AP, LCh, SN, JD, PJ, SM, AL, GFC and TP performed the experiments. LCo, SJ and WM carried out the electrophysiology experiments and analysis. HRRW, FMW and SJDa undertook the UK BioBank association study. SW aided in the management of the *clarinet* colony. SRG, KNA and CP provided support; LAD, CA, LCh, SJDa, SDMB, WM, AE and MRB wrote the manuscript, AE and MRB conceived and coordinated the study.

## Conflict of interest

The authors declare that they have no conflict of interest.

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
