## [Review Process File · EMBO Molecular Medicine]

Clarin-2 is essential for hearing by maintaining stereocilia integrity and function

Lucy A Dunbar, Pranav Patni, Carlos Aguilar, Philomena Mburu, Laura Corns, Helena RR Wells, Sedigheh Delmaghani, Andrew Parker, Stuart Johnson, Debbie Williams, Christopher T Esapa, Michelle M Simon, Lauren Chessum, Sherylanne Newton, Joanne Dorning, Prashanthini Jeyarajan, Susan Morse, Andrea Lelli, Gemma F Codner, Thibault Peineau, Suhasini R Gopal, Kumar N Alagramam, Ronna Hertzano, Didier Dulon, Sara Wells, Frances M Williams, Christine Petit, Sally J Dawson, Steve DM Brown, Walter Marcotti, Aziz El-Amraoui, Michael R Bowl

Review timeline:

Submission date:	9 January 2019
Editorial Decision:	8 February 2019
Revision received:	9 June 2019
Editorial Decision:	11 July 2019
Revision received:	23 July 2019
Accepted:	26 July 2019

Editor: Jingyi Hou

Transaction Report:

1st Editorial Decision

8 February 2019

Thank you for the submission of your manuscript to EMBO Molecular Medicine. We have now heard back from the three referees whom we asked to evaluate your manuscript.

You will see from the comments below that while the referees find the manuscript to be of interest, providing novelty and technical quality, referees #1 and #2 share similar concerns about the weak clinical insights - and referee #1 makes important suggestions to address them. Upon our cross-commenting exercise, referee #3 added "I agree with other referees that the extend of the human results have either to be improved (validation with other cohorts, collaboration with human geneticists collecting deaf families) or toned down. If toned down, it might fall short for reaching the Human pertinence required for your journal." Referee #1 added "I agree that in addition to using the gEAR portal, the expression of *Clrn2* could be experimentally addressed with either in situ hybridization/RNA Scope or single/few cell PCR." We would therefore expect a point-by-point response to all concerns raised by the three referees and accordingly changes in the manuscript text. Importantly, we would like you to test the GERA cohort as referee #1 suggested and strengthen the human correlates. We also expect you to address the mRNA expression pattern of *Clrn2* as referee #1 and #3 suggested.

We would welcome the submission of a revised version within three months for further consideration and would like to encourage you to address all the criticisms raised as suggested to improve conclusiveness and clarity. Please note that EMBO Molecular Medicine strongly supports a single round of revision and that, as acceptance or rejection of the manuscript will depend on another round of review, your responses should be as complete as possible.

I look forward to receiving your revised manuscript.

***** Reviewer's comments *****

Referee #1 (Remarks for Author):

In this study, the authors provide an excellent and comprehensive analysis of two mouse lines with mutations in the CLRN2 gene encoding clarin-2. This protein/gene has not been described before in any scientific publication, and is for the first time demonstrated to be crucial for auditory hair cell function and hearing in mammals. The authors provide immunohistochemical, ultrastructural and functional analyses of inner and outer hair cells, providing a clear image of the role clarin-2 plays for maintenance of hair cell stereocilia. They found that clarin-2 seems to be dispensable for early hair bundle maturation, but in absence of clarin-2, the anchoring of the tip links to the lateral wall of the neighboring larger stereocilia is affected, indicated by mispositioning of harmonin-b immunoreactive spots and a round and oblate shape of the top of middle and short row stereocilia. Subsequently, the lowest row of stereocilia vanishes, both effects leading to a strong reduction in the number of functional mechanotransduction channels. Furthermore, the inner hair cells fail to develop mature K⁺ currents until P22, which might be a downstream effect of the reduced mechanotransduction currents (as discussed by the authors). As a result, inner and outer hair cells are functionally impaired, explaining substantively the profound hearing loss in the mouse lines. The analysis is complemented by histological and functional assessments of vision and behavioral tests for balance deficits, indicating no impairment in CLRN2 mutant mice. The data suggest that pathogenic mutations in the human CLRN2 gene likely also cause profound hearing loss in humans, which I assume might be congenital. Here, the authors found SNPs in or close to the human CLRN2 gene that correlate with late-onset hearing loss. While the strength of this study is clearly the highest level analysis of the inner ear phenotype in the mouse lines, the way of identifying variants in the human gene lacks behind.

Major concern

Given the profound hearing loss in two mouse models, why did the authors suspect that mutations in the human CLRN2 gene will be associated with rather mild forms of age-related hearing loss in first place? Since the hearing impairment in mice is very strong; and already detectable during maturation of the organ of Corti (P8 in OHCs, which is before the onset of hearing in mice), mutations in humans could as well lead to congenital deafness. I assume that such mutations will be identified in the future, once genetic analysis of this gene will be routinely implemented in panel sequencing for hearing impaired individuals, which can be recommended based on this work. However, to date, such mutations in human CLRN2 await their identification, which seemed to force the authors to try to find SNPs in the UK Biobank associated with hearing impairment. While for other deafness genes, e.g. KCNQ4 or TRIOBP, mutations causing early onset hearing impairment but also SNPs associated with age related hearing loss have been identified, the restriction to the latter at least requires some explanation.

A major weak point is that the search for a correlation of hearing impairment and variants in the respective genomic region relies solely on self-reported hearing impairment in the UK Biobank cohort. That way, acquired hearing loss (e.g. by otitis media or sudden hearing loss) or hereditary sensorineural hearing loss cannot be discriminated. A history of noise exposure has obviously not been taken into account, which could have been done to exclude likely cases of noise-induced hearing loss. Moreover, as I understood from the methods, hearing aid users were excluded in case they answered one of the two hearing-related questions with "no". By these settings, individuals with a strong hearing impairment might have been imputed.

Nevertheless, the authors did find an association of impaired hearing with SNPs in or close to the CLRN2 gene, indicating that SNPs in this gene could play a role in age-related hearing loss. A SNP

with clearly pathogenic nature (e.g. inducing a premature STOP codon) has not been found, which might be due to the restriction to late-onset hearing loss cases. From the mouse data provided here, I assume such pathogenic mutations might lead to recessively inherited congenital hearing impairment.

In the Genetic Epidemiology Research on Adult Health and Aging (GERA) cohort, a subgroup has been characterized for hearing impairment by audiologists, indicated by electronic records of the nature of hearing impairment (see Hoffmann et al, PLOS Genetics 2016). Due to this, cases of acquired hearing loss could be excluded. In addition, speech recognition thresholds (SRT) and speech discrimination scores (SDS), as well as a history of noise exposure are reported for a (large) subset of individuals. Could the authors test in this cohort as well for a correlation with SNPs in CLRN2?

Minor points

Page 5

The official sequence variant nomenclature recommends the use the three letter code rather than the one letter code for amino acids, such that the substitution here should be labelled p.Trp4*.

Page 6ff

For all data in numerical format, the decimals should be rounded appropriately, i.e. to full numbers when the standard deviation is >1. (E.g. click thresholds of 63.75 dB SPL {plus minus} 15.06 s.d. should be written as 64 dB SPL {plus minus} 15 s.d.)

Pages 11-12

While describing the K⁺ currents in the "old" physiologist's style (IK,f etc.), could the authors rather provide the names of the respective ion channels (e.g. "BK") whenever known?

Page 14

"post-hearing stages" - presumably, the stage after the onset of hearing is meant (not a stage after hearing loss), please rewrite.

Page 23

Has the liquid junction potential been measured or calculated? Related to that, in Figure 9, you labeled the top traces with -37 mV (in d) but with -34mV (in e), and the second trace label differs by 1 mV between d and e. Did you really apply different protocols here?

Page15

Clarin-1 mutant mice display no visual phenotype, despite humans do when CLRN1 is mutated. Is there any reason why you suggest that no visual impairment in the CLRN2 mouse lines excludes a visual phenotype in CLRN2 mutant humans? Unless you have good reasons, it should at least be discussed that a visual phenotype cannot be fully excluded for human CLRN2 patients. In accordance, the short summary statements, e.g. on page 7, should be complemented with the restriction (regarding absence of retinal deficits) to mice (unless a reason can be given).

Methods/Figure1 (and Figure 2 accordingly)

Auditory testing: since the sound stimulus with highest intensity in the tests was 90dB, had there been animals with no ABR signal at 90dB? If so, please indicate how many of the tested mice did not respond to a 90dB sound/click stimulus. It seems that such mice are displayed at 100dB in Fig. 1a, but it is not described appropriately in the methods or the figure legend. To calculate an average threshold (Fig. 1e), such animals could either be set to 95dB (or 100db), or you calculate the threshold only from the animals with a recognizable ABR.

Figure 5b, right panel

Many of the middle row stereocilia look like they were quite distant from the adjacent larger stereocilia row. Few of the middle row stereocilia, those with less distance to the largest row, seem to be of prolate shape. Is this typical? Do you interpret this as many tip links are missing, but some are present, indicated by the prolate shape? I missed the statement of (likely) missing tip links in most stereocilia and few prolate shaped stereocilia in the results.

Referee #2 (Comments on Novelty/Model System for Author):

The study is nicely done to demonstrate that *Cln2* is involved in hair cell maintenance and normal hearing, and the lack of function leads to hearing loss in mice. The novelty could be greatly increased if the connection between the SNPs with human progressive hearing can be established, which may be too much to ask for the paper.

Referee #2 (Remarks for Author):

It is estimated that there are hundreds of genes in which mutations could lead to genetic hearing loss. Identification and characterization of new deafness genes will help with the understanding of the etiology and provide potential targets for intervention. The study identified *Cln2* as a deafness gene from an ENU mutagenesis project. It provided evidence by morphological, cellular, hearing test and physiological studies that *Cln2* is responsible for genetic hearing loss in mice, as the results of degeneration of stereocilia and the loss of mechano-electrical transduction. It provides limited evidence suggesting *CLRN2* may be involved in human non-syndromic progressive hearing loss.

To improve the papers, the following points need to be addressed.

1. One of the major conclusions is that *CLRN2* is involved in human non-syndromic progressive hearing loss. This is an overstatement without convincing evidence. The conclusion was based on the association study without any experimental evidence. As it is known that association studies could generate false positive in general if they are not supported by experimental data. In the *Cln2* mouse model, the mutation is transmitted as recessive with the loss of function that leads to early hearing loss. In humans, we do not know anything about any effect of the SNPs in the *CLRN2* region. A recent study (Lewis et al, 2018) showed the pitfalls for this type of association study. Without further experiment, the section on human deafness should be significantly toned down. It merely provides some clues for future study.
2. The study did not provide cellular distribution of *Cln2* in the inner ear during development. While the antibody did not work, *in situ* should be done to show when and where the gene is expressed, to help understand its function and inner ear pathology.
3. Disruption of *Xirp2* shows the effect on the stereocilia that resembles the *Cln2* null phenotype. Distribution of *Xirp2* should be studied in the *Cln2*-null hair cells.
4. Page 8, "In addition, the tips of the middle of shortest row stereocilia in *Cln2clarinet/clarinet* mutant IHCs .." is vague, and I do not see the described effect. It should be taken out.
5. Page 8, "However, the tips of the middle of shortest row stereocilia in *Cln2clarinet/clarinet* mice continue to ... (Figure 5b)", the description of a rounded shape is not convincing. They need to provide the measurements to demonstrate the effect. Otherwise it should be avoided.
6. As there are multiple harmonin-b labeling points (punctate) in the *Clarn2* mice, which ones did they count to reflect the distance? The highest one and Why?
7. Page 10, "which is consistent with the round and oblate shape of the stereocilia at this stage in *Cln2clarinet/clarinet* mice (Figure 5b)". To make it true, we have to assume that *CLRN2* is involved in the tension forces and also tension forces determine the shape of the stereocilia to make the assumption. Where is the evidence for the statement?
8. Page 7, "Figure b,c" should be "Figure 3b,c".

Referee #3 (Comments on Novelty/Model System for Author):

Review paper Dunbar et al.

Clarin-2 is essential for hearing by maintaining stereocilia integrity and function.

In this paper, the authors characterize a new deafness gene, *Clarin-2*, from a forward genetic screen done in mice. They named this mouse mutant *Clarinet*. By generating a second mouse model with targeted deletion, they prove by complementation test that it is the causal gene.

The study takes advantage of the Biobank database to interrogate the 100Kb around human *CLARN2* with hearing impaired traits. They remarkably found a cluster of 5 SNPs. Detailed analysis of the SNPs with the highest association lie 2 Kb downstream of *CLARN2* and a non-synonymous Leu to Val change within the predicted second transmembrane domain.

The authors performed a physiology scan for hearing, vestibular and visual functions in the *Clarinet* mutant. They found that only the hearing function was affected (non-syndromic HL), with early-onset and progressive nature.

Then they inspect the stereocilia bundle of cochlear hair cells and found morphological defect starting after the first post-natal week: they found a reduction of height affecting more the smaller rows in OHC and a rounded-shape at the tips of transducing rows, previously linked with tip link loss. Injectoporation of GFP-*Clarin-2* shows a localization in the stereocilia bundle. The authors then investigate if PDZ-containing proteins present in the bundle could be mislocalized in the *Clarin-2* mutant. They looked at *Whirlin*, *PDZD7*, and *Harmonin*. They conclude that *Whirlin* and *PDZD7* are not affected (but see comment below). They found and quantified the mislocalization of *Harmonin* from the upper insertion point of the TL to high heights in the stereocilia, likely resulting from the TL disruption. They evaluate the mechanotransducer currents by fluid-jet and found a decrease in amplitude but not of resting P_o . They also investigate the maturation of the basolateral current of the OHC and IHC. They found no changes for the OHC while the IHC kept they pre-hearing characteristics (I_{k1} present, $I_{k,n}$, and $I_{k,f}$ absent).

I found this study very comprehensive for the first characterization of a new deafness gene and well led. The main results are that both human and mice are affected by a non-syndromic form of HL, and that *Clarin-2* is likely involved in the TL maintenance starting at P6.

Of course, the study would gain by having a protein localization, but this is a common problem in the field, and the authors tried a bundle of strategies without success. It would also have been informative to count tip links to determine if there is a direct correspondence between numbers of TL and max current.

major concerns that would need to be addressed.

- Even if the protein localization is not working, it would be useful for the reader to visualize the mRNA expression pattern of *Clarin-2* in the cochlea and the vestibule. Getting the information from gEAR portal would be one way to do this, the best one being performing in situ hybridization. This would be particularly important as the expression information of gEAR portal show HC specific expression at P1 only. Maybe later other cells are also expressing *Clarin-2*.

- In fig.7a, the authors immune-localized *PDZD7*, an ankle link component, in WT and *Clarinet* cochlear HC at P6. They conclude that "the subcellular distribution of *whirlin* and *PDZD7* were identical in *Clrn2clarinet/clarinet* mice and *Clrn2clarinet/+* littermates". However, when I look at the IHC localization in the *Clarinet* mutant at P6 (in Fig 7a and Supp 7b), it seems different than littermate controls. I could not find information in the paper about the origin and specificity of the anti-*PDZD7* used; this should be corrected. The localization presented herein control at two levels, one might be the ankle link area of the tallest row, but it is difficult to state on the second level of the signal located in lower rows. In the *Clarinet* mutant, the *PDZD7* staining is present as a single level, in the tallest row. Is this a phenotype? More experiment to clarify this point is needed.

- There is a confusion in the use of Upper Tip-Link insertion point and Upper Tip-Link Density (UTLD) terms. To the best of my knowledge, the UTLD density has never been showed before P10 and consequently can not be used for P6 upper TL insertion point. Therefore the Fig.7c, legend, and methods have to be corrected

- More information about the *Clrn2* del629 is needed. What is the consequence of the deletion? A frameshift? Is there a residual protein putatively produced? Looking at the mRNA in the mutant would be essential to define this point.

- In the discussion, the nature and position of the human *CLRN2* variants identified in HL patients are not discussed. This is missing.

minor points

- Concerning the quantification of Harmonin Spot along the tallest row, the sentence in the text lack the reference point (relative to the tip of the taller stereocilium) for the measurements: "The harmonin-b immunoreactive spots were observed on average at 575.4 {plus minus} 23.15 nm (mean {plus minus} s.e.m.; n=35 hair bundles from 5 mice) in *Clrn2*clarinet/clarinet mice, as compared to 850 {plus minus} 28.2 nm (mean {plus minus} s.e.m.; n=31 hair bundles from 5 mice) in *Clrn2*clarinet/+ mice (Figure 7f) (p>0.0001, student's t-test)."

- The gene nomenclature changed the gene name of *TMHS* to *LHFPHL5*. The HUGO gene nomenclature Committee still indicated it should be *LHFPL5*, and therefore this should be corrected.

1st Revision - authors' response

9 June 2019

Thank you very much for the opportunity to revise our manuscript for publication in EMBO Molecular Medicine. Below is a point-by-point response to the Referees comments. We believe that we have managed to address all of the concerns raised.

Referee #1:

We thank Referee 1 for their comment - "*In this study, the authors provide an excellent and comprehensive analysis of two mouse lines with mutations in the CLRN2 gene encoding clarin-2*". Moreover, we have addressed their concerns as listed below.

Major concern:

1. *Given the profound hearing loss in two mouse models, why did the authors suspect that mutations in the human CLRN2 gene will be associated with rather mild forms of age-related hearing loss in first place? Since the hearing impairment in mice is very strong; and already detectable during maturation of the organ of Corti (P8 in OHCs, which is before the onset of hearing in mice), mutations in humans could as well lead to congenital deafness. I assume that such mutations will be identified in the future, once genetic analysis of this gene will be routinely implemented in panel sequencing for hearing impaired individuals, which can be recommended based on this work. However, to date, such mutations in human CLRN2 await their identification, which seemed to force the authors to try to find SNPs in the UK Biobank associated with hearing impairment. While for other deafness genes, e.g. *KCNQ4* or *TRIOBP*, mutations causing early onset hearing impairment but also SNPs associated with age related hearing loss have been identified, the restriction to the latter at least requires some explanation.*

Response: Clarin-1 total knockout mice display profound hearing loss already by postnatal day 15, whilst disabling causal mutations in humans cause post-lingual progressive hearing loss in Usher syndrome type III (USH3) patients. Indeed, almost all USH3A patients develop normal speech, and, although an elevation of hearing thresholds is diagnosed in most patients before the age of 10 years, some patients display only mild-to-moderate hearing threshold elevation at the time of detection, at an age of 30 to 40 years (Pakarinen et al. 1995; Ness et al. 2003). Moreover, the *Clrn1*^{N48K} mouse mutant exhibits profound hearing loss by P25 (Geng et al, 2012), even though USH3a patients with the *CLRN1*^{N48K} mutation display post-lingual progressive hearing loss (Ness et al, 2003). Our findings in *clarinet* mice show that unlike clarin-1, which is required during embryonic stages, clarin-2 is dispensable for the patterning and establishment of the 'staircase' bundle in young postnatal hair cells. Thus, our hypothesis that clarin-2 has an important role in functionally mature cochlear hair cells, and that potential *CLRN2* mutations also might cause late-onset hearing impairment in humans (see Discussion, page 15). For this reason, we investigated whether variation(s) in this gene was associated with human hearing in the UK Biobank cohort (the largest population cohort available with 87,056 cases and 163,333 controls).

Pakarinen L, Karjalainen S, Simola KO, Laippala P, Kaitalo H. Usher's syndrome type 3 in Finland. *Laryngoscope*. 1995 Jun;105(6):613-7.

Ness SL, Ben-Yosef T, Bar-Lev A, Madeo AC, Brewer CC, Avraham KB, Kornreich R, Desnick RJ, Willner JP, Friedman TB, et al. Genetic homogeneity and phenotypic variability among Ashkenazi Jews with Usher syndrome type III. *J Med Genet.* 2003;40(10):767-72.

Geng R, Melki S, Chen DH, Tian G, Furness DN, Oshima-Takago T, Neef J, Moser T, Askew C, Horwitz G, et al: The mechanosensory structure of the hair cell requires clarin-1, a protein encoded by Usher syndrome III causative gene. *J Neurosci* 2012, 32:9485-9498.

2. *A major weak point is that the search for a correlation of hearing impairment and variants in the respective genomic region relies solely on self-reported hearing impairment in the UK Biobank cohort. That way, acquired hearing loss (e.g. by otitis media or sudden hearing loss) or hereditary sensorineural hearing loss cannot be discriminated. A history of noise exposure has obviously not been taken into account, which could have been done to exclude likely cases of noise-induced hearing loss. Moreover, as I understood from the methods, hearing aid users were excluded in case they answered one of the two hearing-related questions with "no". By these settings, individuals with a strong hearing impairment might have been imputed.*

Response: It is correct that in the UK Biobank cohort individuals are defined by self-reported hearing impairment, as it is extremely difficult to perform audiometric assessments on such a large scale (87,056 cases and 163,333 controls). However, such an extremely large sample size provides the greater power needed to detect genetic effects and avoids the sampling bias of studies utilising small cohorts that do have pure tone audiogram data, which are prone to false positive associations. Indeed, this has been the experience of previous studies of genetic associations with age-related hearing loss where sample sizes are much smaller, generally a few thousand cases and none with more than 7,000 cases. The large sample size used here makes our finding much more robust.

We cannot exclude that the sample may include individuals that have otitis media, sudden hearing loss, noise-induced hearing loss or congenital forms of hearing loss. However, our aim was to assess whether *CLRN2* plays a role in risk of common forms of late-onset hearing loss, as such we limited the minor allele frequency to 0.01 which will exclude rare, highly pathogenic variants causing congenital deafness and excluded participants who were completely deaf.

In addition, individuals were excluded if they responded “no” to both hearing difficulty questions “Do you have any difficulty with your hearing?” and “Do you find it difficult to follow a conversation if there is background noise (such as TV, radio, children playing)?”, but responded “Yes” to “Do you use a hearing aid most of the time”. This combination of responses implies that an individual has no difficulty with their hearing or hearing in background noise, and yet has been prescribed a hearing aid. Such individuals were omitted as their responses are contradictory, and therefore were deemed unreliable. Moreover, the presence of any individuals with otitis media and sudden hearing loss within the cohort would be expected to reduce the power to detect an association, therefore the fact that we still detect a highly significant association is notable.

3. *Nevertheless, the authors did find an association of impaired hearing with SNPs in or close to the CLRN2 gene, indicating that SNPs in this gene could play a role in age-related hearing loss. A SNP with clearly pathogenic nature (e.g. inducing a premature STOP codon) has not been found, which might be due to the restriction to late-onset hearing loss cases. From the mouse data provided here, I assume such pathogenic mutations might lead to recessively inherited congenital hearing impairment.*

Response: We agree with the possibility that strongly pathogenic *CLRN2* mutations (e.g. nonsense alleles) might lead to recessively inherited severe hearing impairment. However, as has been demonstrated for *CLRN1* (see above point 1), age of onset of hearing impairment might differ between mice and humans. Thus, it is likely that depending on the severity of a given *CLRN2* mutation, patients might display early or late onset, mild or severe hearing loss. So far, within our teams, whole exome sequencing has been performed on a few hundred patients, mostly of Mediterranean origin, who exhibit congenital or early-onset, profound deafness. To date, this has not revealed any potential *CLRN2* mutation. However, a collaborative work between the teams of Dr. Suhasini R. Gopal (corresponding author in the JCI article) and Pr. K. Alagramam (Case Western Reserve University, USA), Dr Barbara Vona (Julius Maximilians University of Würzburg Germany), Pr. Hamid Galehdari (Shahid Chamran University of Ahvaz, Iran), Pr. Gholamreza Shariati (Ahaz Jundishapur University of Medical Sciences, Iran), and Pr R. Smith (University of Iowa) recently reported an novel missense variant, c.494C>A (p.Thr165Lys) in *CLRN2*, in 3 patients (29, 44, and 25 years old) in an extended consanguineous Iranian family segregating autosomal

recessive non-syndromic sensorineural hearing loss. These patients develop postlingual, moderate-to-profound, bilateral autosomal recessive non-syndromic sensorineural hearing loss. (Gopal et al. ARO abstract PD 84, p520. 2019*). When the mutant $Clrn2^{T165K}$ -YFP protein was expressed in zebrafish hair cells, it failed to localize to the hair bundle or plasma membrane. In addition, *in silico* and *in vitro* analyses using mini-gene assays revealed defective splicing and a shift in the reading frame as a result of this variant. This work, which supports a key role for clarin-2 in human hearing, is now referred to in our manuscript (Page 16).

*This work has been submitted to *Journal of Clinical Investigation*, and is currently under revision: Gopal SR, Vona B, Azaiez H, Mazaheri N, Booth KT, Maroofian R, Clancy K, Shariati G, Sedaghat A, Stepanyan R, Smith RJH, Haaf T, Galehdari H, Alagramam KN. **Mutations in *CLRN2* cause hearing loss in human and a zebrafish model reveals possible etiology.**

4. *In the Genetic Epidemiology Research on Adult Health and Aging (GERA) cohort, a subgroup has been characterized for hearing impairment by audiologists, indicated by electronic records of the nature of hearing impairment (see Hoffmann et al, PLOS Genetics 2016). Due to this, cases of acquired hearing loss could be excluded. In addition, speech recognition thresholds (SRT) and speech discrimination scores (SDS), as well as a history of noise exposure are reported for a (large) subset of individuals. Could the authors test in this cohort as well for a correlation with SNPs in *CLRN2*?*

Response: Replication in a separate cohort, showing a correlation with SNPs in *CLRN2* and late-onset human hearing impairment, would indeed provide a validation of our finding. However, the difficulty is the lack of available replication cohorts with large enough sample sizes to provide sufficient power. GERA is the second largest cohort available after the UKBB cohort, but even this cohort has only 6,527 cases, an order of magnitude 10x smaller than the UKBB cohort that has 87,056 cases. GERA does have 45,882 controls, but an excess of controls does not significantly improve statistical power in genetic association studies. We have performed a power calculation (Purcell et al. 2003) to assess the power of the GERA cohort to replicate our finding and found only 56% power to replicate the strongest *CLRN2* association ($p < 0.05$, additive model, allelic effect). These calculations suggest that 11,378 cases would be required for 80% power, which is the minimum level considered for genetic association studies.

Purcell S, Cherny SS, Sham PC. (2003) Genetic Power Calculator: design of linkage and association genetic mapping studies of complex traits. *Bioinformatics*, 19(1):149-150.

Minor comments:

5. *Page 5 - The official sequence variant nomenclature recommends the use the three letter code rather than the one letter code for amino acids, such that the substitution here should be labelled p.Trp4*.*

Response: This has been changed on page 5, in the legend to Figure 1, and in the legend to Figure EV1.

6. *Page 6 - For all data in numerical format, the decimals should be rounded appropriately, i.e. to full numbers when the standard deviation is >1. (E.g. click thresholds of 63.75 dB SPL {plus minus} 15.06 s.d. should be written as 64 dB SPL {plus minus} 15 s.d.).*

Response: These changes have been made.

7. *Pages 11-12 - While describing the K^+ currents in the "old" physiologist's style ($I_{K,f}$ etc.), could the authors rather provide the names of the respective ion channels (e.g. "BK") whenever known?*

Response: We have added the names of the ion channels that have been identified e.g. SK2 and BK (Page 12). The channels carrying I_K , I_{K1} and I_{Na} currents are not known.

8. *Page 14 - "post-hearing stages" - presumably, the stage after the onset of hearing is meant (not a stage after hearing loss), please rewrite.*

Response: We have edited the sentence to now read ". . . indicating a potential key function after the onset of hearing (~P12 in mice)." (Page 13).

9. *Page 23 - Has the liquid junction potential been measured or calculated? Related to that, in Figure 9, you labeled the top traces with -37 mV (in d) but with -34mV (in e), and the second trace label differs by 1 mV between d and e. Did you really apply different protocols here?*

Response: We have now specified that the liquid junction potential (LJP) has been measured between electrode and bath solutions (Page 25). Regarding Figure 9 d and e, the protocol used is the same (as specified in the legend). The difference is because, as mentioned in the Method section, the indicated V_m is corrected not only for the LJP but also for the voltage drop across the residual series resistance after compensation, which can vary between cells. Although this procedure is very often ignored in the field, we believe that it is crucial to provide a “true” representation of the voltage reached by the cells during voltage clamp.

10. *Page15 - Clarin-1 mutant mice display no visual phenotype, despite humans do when CLRN1 is mutated. Is there any reason why you suggest that no visual impairment in the CLRN2 mouse lines excludes a visual phenotype in CLRN2 mutant humans? Unless you have good reasons, it should at least be discussed that a visual phenotype cannot be fully excluded for human CLRN2 patients. In accordance, the short summary statements, e.g. on page 7, should be complemented with the restriction (regarding absence of retinal deficits) to mice (unless a reason can be given).*

Response: As mentioned above, Gopal *et al* (Gopal *et al*. ARO abstract PD 84, p520. 2019) have recently reported autosomal recessive non-syndromic sensorineural hearing loss in a consanguineous Iranian family caused by a *CLRN2* gene mutation (p.Thr165Lys). When the mutant *Clrn2*^{T165K}-YFP protein was transiently expressed in zebrafish hair cells, it failed to localize to the hair bundle or plasma membrane. In addition, *in silico* and *in vitro* analyses using mini-gene assays revealed defective splicing and a shift in the reading frame as a result of this variant. The three probands, aged 25, 29 and 44 years old, exhibit profound hearing loss with no indication of vision or balance deficits. Thus, while mutations in *CLRN1* can lead to Usher (Deaf-Blindness) syndrome, all current data suggest that *CLRN2* mutations most likely causes non-syndromic hearing loss. This part of the text has been modified to include these new information (Page 16).

11. *Methods/Figure1 (and Figure 2 accordingly) - Auditory testing: since the sound stimulus with highest intensity in the tests was 90dB, had there been animals with no ABR signal at 90dB? If so, please indicate how many of the tested mice did not respond to a 90dB sound/click stimulus. It seems that such mice are displayed at 100dB in Fig. 1a, but it is not described appropriately in the methods or the figure legend. To calculate an average threshold (Fig. 1e), such animals could either be set to 95dB (or 100db), or you calculate the threshold only from the animals with a recognizable ABR.*

Response: For *clarinet* ABRs, we recorded the threshold of mice not showing a response at the maximum level tested (90 dB SPL) as 95 dB SPL. These mice/thresholds were included when calculating the average threshold. As such, Figure 1e and 2b-e have not changed. However, for consistency we have modified Figure 1a, which shows MPC 169 ABR thresholds, to be consistent with the ‘no response’ set point. We have also added text (see below) to the ‘Auditory phenotyping’ paragraph in the Methods section and, in the figure legends we have indicated the number of mice exhibiting no ABR response at 90 dB SPL.

In the Methods section (page 18) we have added:

For graphical representation, mice not showing an ABR response at the maximum level tested (90 dB SPL) were recorded as having a threshold of 95 dB SPL. These mice/thresholds were included when calculating genotype average thresholds.

In the legend to Figure 1 (page 32) we have added:

Indeed, all eight affected mice were found to not respond to the highest intensity stimulus (90 dB SPL) at the three frequencies tested, or the click stimulus, and so their thresholds are shown as 95 dB SPL.

And:

*All five *Clrn2*^{clarinet/del629} mice were found to not respond at the highest intensity stimulus (90 dB SPL) for at least one frequency-specific /click stimulus.*

In the legend to Figure 2 (page 34) we have added:

*At P16, all eight *Clrn2*^{clarinet/clarinet} mice exhibited recordable ABR responses for each frequency tested and click stimulus. For the longitudinal ABR study, at P21 and P28 three of the seven *Clrn2*^{clarinet/clarinet} mice*

were found to not respond at the highest intensity stimulus (90 dB SPL) for at least one frequency/click stimulus. By P42, five of the *Clrn2*^{clarinet/clarinet} mice were found to not respond at the highest intensity stimulus (90 dB SPL) for at least two frequency-specific/click stimuli.

12. *Figure 5b, right panel - Many of the middle row stereocilia look like they were quite distant from the adjacent larger stereocilia row. Few of the middle row stereocilia, those with less distance to the largest row, seem to be of prolate shape. Is this typical? Do you interpret this as many tip links are missing, but some are present, indicated by the prolate shape? I missed the statement of (likely) missing tip links in most stereocilia and few prolate shaped stereocilia in the results.*

Response: The preserved tenting at the extreme tip of the transducing stereocilia (the short and middle rows) is indicative of persisting tension (in series with the tip link) due to mechano-electrical transduction activity. More extensive morphological studies, at mature ages, would be necessary to accurately determine the presence/absence of tip links in control and mutant mice. Indeed, the preservation of the tip-links, even in wild-type conditions, is extremely challenging. Nonetheless, apart from morphological analyses, and based on our MET current data measured at P7, we believe that at this stage the tip links do form in absence of clarin-2. The loss in some stereocilia of the prolate (elongated) shape might be due to reduced tension, despite still persisting tip links. This can occur due to reduced coupling between the stereociliary membrane and F-actin, which is consistent with the identified loss of restricted localization of harmonin-b in mutant stereocilia, a protein that binds to both the tip-link component cadherin-23 and to actin filaments.

Referee #2:

We thank Referee 2 for their comment - *“The study is nicely done to demonstrate that *Clrn2* is involved in hair cell maintenance and normal hearing, and the lack of function leads to hearing loss in mice”*. Moreover, we have addressed their concerns with additional experiments, analyses and clarifications as listed below.

Major concerns:

1. *One of the major conclusions is that *CLRN2* is involved in human non-syndromic progressive hearing loss. This is an overstatement without convincing evidence. The conclusion was based on the association study without any experimental evidence. As it is known that association studies could generate false positive in general if they are not supported by experimental data. In the *Clrn2* mouse model, the mutation is transmitted as recessive with the loss of function that leads to early hearing loss. In humans, we do not know anything about any effect of the SNPs in the *CLRN2* region. A recent study (Lewis et al, 2018) showed the pitfalls for this type of association study. Without further experiment, the section on human deafness should be significantly toned down. It merely provides some clues for future study.*

Response: Please see our response to Referee 1 (point 3), which describes a recently reported consanguineous Iranian family suffering autosomal recessive non-syndromic sensorineural hearing loss due to a missense mutation in the *CLRN2* gene (Gopal et al. ARO abstract PD 84, p520. 2019). This new research is now referred to in our manuscript (Page 16).

Furthermore, although it is correct that candidate gene association studies can provide false associations there are a number of factors that make our observation more robust:

- The large size of the cohort, which is more than 10 times larger than previous studies (87,056 cases).
- The strength and number of SNP associations at this locus.
- The phenotype of the mouse is compatible with a role in maintenance of hearing.

The Lewis paper (2018) used a pilot sample whole exome sequencing of 30 individuals with hearing loss and highlighted the danger of comparing these predicted pathogenic mutations to data from unscreened controls in population databases such as EXAC and 1000 Genomes, as a large number of people in these cohorts carry predicted pathogenic mutations for hearing loss. This study avoids those pitfalls by using a very large sample size and using controls from the same cohort who report no problems with hearing loss.

Moreover, we have based our research studies on the current knowledge that exists regarding Clarin-1 and human hearing function (commented on in point 1, referee 1). Namely, while the mouse *Clrn1*^{N48K} model exhibits profound hearing loss by P25 (Geng et al, 2012), patients carrying the *CLRN1*^{N48K} allele show

progressive hearing loss with onset between 3- and >35-years of age (average onset is 10-years) (Ness et al, 2003). As such, this prompted us to screen for association between *CLRN2* and adult hearing difficulty in the UK biobank population cohort. This has identified a cluster of SNPs in, and around, the *CLRN2* gene locus that are highly associated with adult hearing difficulty. We are not suggesting these SNPs are directly causing hearing difficulty in this cohort, it is more likely that these SNPs are in linkage disequilibrium with an as yet unidentified causal variant(s). However, one of the SNPs does lie within the *CLRN2* coding sequence, with presence of the minor allele causing a missense substitution of an evolutionarily conserved Leucine residue (p.Leu113Val). Similar Leu to Val substitutions, also located in the highly conserved regions of the protein transmembrane domains, encoded by, have been reported in the gene *PSENI* in patients with Alzheimer's disease, supporting their potential pathogenicity (see text, page 15). However, a substantial amount of additional work would be required to ascertain the effect of the p.Leu113Val missense variant on hearing function, likely through the generation and characterization of a knock-in mouse mutant. Nonetheless, to validate the involvement of a gene, identified through human association studies, in disease causation has historically involved the generation of a knockout mouse model, which in essence is what the *clarinet* (and *del629*) mutant provides.

Thus, we believe that our combined data set from mice and humans, when taken together, represents the experimental validation the reviewer has requested. Furthermore, the newly reported human mutation data, support our hypothesis that milder mutations of *CLRN2* might predispose to progressive, late-onset hearing loss.

Geng R, Melki S, Chen DH, Tian G, Furness DN, Oshima-Takago T, Neef J, Moser T, Askew C, Horwitz G, et al: The mechanosensory structure of the hair cell requires clarin-1, a protein encoded by Usher syndrome III causative gene. *J Neurosci* 2012, 32:9485-9498.

Ness SL, Ben-Yosef T, Bar-Lev A, Madeo AC, Brewer CC, Avraham KB, Kornreich R, Desnick RJ, Willner JP, Friedman TB, et al. Genetic homogeneity and phenotypic variability among Ashkenazi Jews with Usher syndrome type III. *J Med Genet.* 2003;40(10):767-72.

2. *The study did not provide cellular distribution of Clnr2 in the inner ear during development. While the antibody did not work, in situ should be done to show when and where the gene is expressed, to help understand its function and inner ear pathology.*

Response: We analyzed the expression of *Clnr2* in the inner ear at postnatal and adult stages using the gEAR portal (umgear.org). Consistent with our RT-PCR findings at embryonic (E17.5) and postnatal (P4, P8, P12, P16 & P28; see Figure EV4a) timepoints, expression datasets reveal that while the *Clnr2* transcript is lowly expressed in the newborn inner ear and early postnatal stages, it is readily detected in P15 and adult sorted hair cells (Liu et al. 2014; Liu et al. 2018; Ranum et al. 2019). Interestingly, *Clnr2* transcripts were detected also in the auditory cortex (A1) and increased in levels between P7 to adult mice (Guo et al. 2016). Moreover, undertaking RT-PCR using RNA extracted from inner and outer hair cells isolated from P15 wild type mice, we could also confirm the expression of *Clnr2* transcripts in hair cells (Figure 2b). Consistently, our phenotypic characterization of the clarin-2 deficient mice clearly show significant morphological and functional abnormalities restricted to the auditory hair cells.

Liu H, Pecka JL, Zhang Q, Soukup GA, Beisel KW, He DZ: Characterization of transcriptomes of cochlear inner and outer hair cells. *J Neurosci* 2014, 34:11085-11095.

Liu H, Chen L, Giffen KP, Stringham ST, Li Y, Judge PD, Beisel KW, He DZZ: Cell-Specific Transcriptome Analysis Shows That Adult Pillar and Deiters' Cells Express Genes Encoding Machinery for Specializations of Cochlear Hair Cells. *Front Mol Neurosci* 2018, 11:356.

Ranum PT, Goodwin AT, Yoshimura H, Kolbe DL, Walls WD, Koh JY, He DZZ, Smith RJH: Insights into the Biology of Hearing and Deafness Revealed by Single-Cell RNA Sequencing. *Cell Rep* 2019, 26:3160-3171.e3163.

Guo Y, Zhang P, Sheng Q, Zhao S, Hackett TA: lncRNA expression in the auditory forebrain during postnatal development. *Gene* 2016, 593:201-216.

Minor comments:

1. *Disruption of Xirp2 shows the effect on the stereocilia that resembles the Clnr2 null phenotype. Distribution of Xirp2 should be studied in the Clnr2-null hair cells.*

Response:

A role for Xirp2 in protecting F-actin against depolymerization has been proposed (Scheffer et al. 2015, Francis et al. 2015), and surviving adult mutant mice lacking Xirp2 have been shown to display progressive moderate high-frequency hearing Loss (Francis et al. 2015). The cross-linking and maintenance of F-actin are crucial to stereocilia; as such these functions are secured by redundancy among F-actin associated proteins. However, in the absence of clarin-2, the resultant hearing loss covers all frequencies, and rapidly progresses to profound deafness, indicating it affects hair bundles throughout the cochlea (our study). The more severe *Clrn2*-mediated phenotype probably is the result of an ionic homeostasis imbalance as a consequence of defects in the mechano-electrical transduction machinery. This in turn also impacts functioning of most stereocilia core proteins, including F-actin binding proteins and membrane to cytoskeleton connectors, such as Xirp2 and PDZD7. As discussed in our manuscript (Page 14), this would explain the similarity in phenotype (regression of short transducing stereocilia) between *Clrn2* mice, and mutant mice lacking any of other key components of the MET machinery. As we move forward with our studies of *Clrn2*, we will look to assess the localization of several bundle-related proteins, including Xirp2. However, obtaining reliable, commercially available antibodies for these proteins is not straightforward. Indeed, the Xin β (D-18): sc-83128 antibody from Santa-cruz, which has been reported to label stereocilia *in situ*, is unfortunately no longer available for order.

Scheffer DI, Zhang DS, Shen J, Indzhykulian A, Karavitaki KD, Xu YJ, Wang Q, Lin JJ, Chen ZY, Corey DP. Xirp2, an actin-binding protein essential for inner ear hair-cell stereocilia. *Cell Rep*. 2015 Mar 24;10(11):1811-8. doi: 10.1016/j.celrep.2015.02.042.

Francis SP, Krey JF, Krystofiak ES, Cui R, Nanda S, Xu W, Kachar B, Barr-Gillespie PG, Shin JB. A short splice form of Xin-actin binding repeat containing 2 (XIRP2) lacking the Xin repeats is required for maintenance of stereocilia morphology and hearing function. *J Neurosci*. 2015 Feb 4;35(5):1999-2014. doi: 10.1523/JNEUROSCI.3449-14.2015.

2. *Page 8, "In addition, the tips of the middle of shortest row stereocilia in *Clrn2*^{clarinet/clarinet} mutant IHCs .." is vague, and I do not see the described effect. It should be taken out.*

&

3. *Page 8, "However, the tips of the middle of shortest row stereocilia in *Clrn2*clarinet/clarinet mice continue to ... (Figure 5b)", the description of a rounded shape is not convincing. They need to provide the measurements to demonstrate the effect. Otherwise it should be avoided.*

Response: As discussed above (point 12, Referee 1), the preserved tenting at the extreme tip of the transducing stereocilia (the short and middle rows) (prolate shape) is indicative of persisting tension (in series with the tip link) due to mechano-electrical transduction activity. The loss of the prolate shape in some stereocilia of clarin-2 deficient hair bundles might be due to tension weakening. We used a quantitative approach (Pages 8 and 22, and Figure EV3) to score prolateness of the second stereocilia row in both inner and outer hair cells, using cochlear mid-turn electron micrographs from control and clarin-2 deficient hair bundles at P8. At least 80 tip images per genotype were used - 3 animals per genotype, 3 bundles per animal. We found a high prevalence of a rounded shape of the second tallest row of stereocilia from *clarinet* mice, as compared to age-matched wild type mice, where a prolate shape is far more common ($p < 0.005$ for all cases, χ^2).

4. *As there are multiple harmonin-b labeling points (punctate) in the *Clrn2* mice, which ones did they count to reflect the distance? The highest one and Why?*

Response: As indicated in the Methods section (Page 22, 1st paragraph), to quantify the positioning of the harmonin-b immunoreactive area relative to the tip of the taller stereocilium (indicative of the upper attachment to the tip link), we focused on the tallest stereocilia in the IHC hair bundle. This offers an unobstructed view of the distal region of individual stereocilia, including the tip and the harmonin-b immunoreactive spot for each stereocilium. Thus, allowing an accurate measurement of the distance of the harmonin-b immunoreactive spot from the stereocilium tip (by ImageJ software (NIH), as indicated in insets, Fig. 7e). Furthermore, measurements were made using IHC, rather than OHC, bundles as it is easier to define individual stereocilia. Confocal images of 8-10 well-preserved IHC bundles, obtained from 4 *clarinet* and 4 wild-type mice were used. For each bundle, the distance from the stereocilium tip to the center of the harmonin-b-immunoreactive area (present on that particular stereocilium) was measured

(by ImageJ software, NIH) for the 3 tallest stereocilia and averaged. Obtained data were analyzed using student's t-test.

5. *Page 10, "which is consistent with the round and oblate shape of the stereocilia at this stage in $Clrn2^{clarinet/clarinet}$ mice (Figure 5b)". To make it true, we have to assume that CLRN2 is involved in the tension forces and also tension forces determine the shape of the stereocilia to make the assumption. Where is the evidence for the statement?*

Response: During hair bundle morphogenesis, the tips of the 'transducing' (short and medium) stereocilia evolve from a round, oblate shape into an asymmetric, prolate shape in wild-type hair bundles, a change ascribed to tension forces applied via the tip link to the apical membrane of these stereocilia (Rzadzinska et al. 2004). Such tenting has since been considered as an indicator of membrane tension, which might be favorable for activating the MET channel (see Sakaguchi et al. 2009, and references therein). The occurrence of some round/oblate stereociliary tips in the absence of Clarin-2 might therefore suggest that these putative tension forces may not develop properly in the *clarinet* mutant hair bundles. As indicated above, points 2 & 3, lack of clarin-2 clearly leads to reduced MET activity, which also manifest by a significant loss of prolateness of the tips of the transducing stereocilia. The molecular mechanisms leading to tension weakening remains to be established.

Rzadzinska AK, Schneider ME, Davies C, Riordan GP, Kachar B: An actin molecular treadmill and myosins maintain stereocilia functional architecture and self-renewal. *J Cell Biol* 2004, 164:887-897.

Sakaguchi H, Tokita J, Müller U, Kachar B. Tip links in hair cells: molecular composition and role in hearing loss. *Curr Opin Otolaryngol Head Neck Surg.* 2009 Oct;17(5):388-93.

6. *Page 7, "Figure b,c" should be "Figure 3b,c".*

Response: This has been corrected.

Referee #3:

We thank Referee 3 for their comment - "I found this study very comprehensive for the first characterization of a new deafness gene and well led". Moreover, we have addressed their concerns as listed below.

Major concerns:

1. *Even if the protein localization is not working, it would be useful for the reader to visualize the mRNA expression pattern of Clarin-2 in the cochlea and the vestibule. Getting the information from gEAR portal would be one way to do this, the best one being performing in situ hybridization. This would be particularly important as the expression information of gEAR portal show HC specific expression at P1 only. Maybe later other cells are also expressing Clarin-2.*

Response: As indicated above (point 2, referee 2), we analyzed the expression of *Clrn2* in the inner ear at postnatal and adult stages using the gEAR portal (umgear.org). Consistent with our RT-PCR findings at embryonic (E17.5) and postnatal (P4, P8, P12, P16 & P28; see Figure EV4a), expression datasets reveal that while the *Clrn2* transcript is lowly expressed in the newborn inner ear and early postnatal stages, it is readily detected in P15 and adult sorted hair cells (Liu et al. 2014; Liu et al. 2018; Ranum et al. 2019). Interestingly, *Clrn2* transcripts were detected also in the auditory cortex (A1) and increased in levels between P7 to adult mice (Guo et al. 2016). Moreover, undertaking RT-PCR using RNA extracted from inner and outer hair cells isolated from P15 wild type mice, we could also confirm the expression of *Clrn2* transcripts in the hair cells (Figure 2b). Consistently, our phenotypic characterization of the Clarin-2 deficient mice clearly show significant morphological and functional abnormalities restricted to the auditory hair cells.

Liu H, Pecka JL, Zhang Q, Soukup GA, Beisel KW, He DZ: Characterization of transcriptomes of cochlear inner and outer hair cells. *J Neurosci* 2014, 34:11085-11095.

Liu H, Chen L, Giffen KP, Stringham ST, Li Y, Judge PD, Beisel KW, He DZZ: Cell-Specific Transcriptome Analysis Shows That Adult Pillar and Deiters' Cells Express Genes Encoding Machinery for Specializations of Cochlear Hair Cells. *Front Mol Neurosci* 2018, 11:356.

Ranum PT, Goodwin AT, Yoshimura H, Kolbe DL, Walls WD, Koh JY, He DZZ, Smith RJH: Insights into the Biology of Hearing and Deafness Revealed by Single-Cell RNA Sequencing. *Cell Rep* 2019, 26:3160-3171.e3163.

Guo Y, Zhang P, Sheng Q, Zhao S, Hackett TA: lncRNA expression in the auditory forebrain during postnatal development. *Gene* 2016, 593:201-216.

2. *In fig. 7a, the authors immune-localized PDZD7, an ankle link component, in WT and Clarinet cochlear HC at P6. They conclude that "the subcellular distribution of whirlin and PDZD7 were identical in Clrn2clarinet/clarinet mice and Clrn2clarinet/+ littermates". However, when I look at the IHC localization in the Clarinet mutant at P6 (in Fig 7a and Supp 7b), it seems different than littermate controls. I could not find information in the paper about the origin and specificity of the anti-PDZD7 used; this should be corrected. The localization presented herein control at two levels, one might be the ankle link area of the tallest row, but it is difficult to state on the second level of the signal located in lower rows. In the Clarinet mutant, the PDZD7 staining is present as a single level, in the tallest row. Is this a phenotype? More experiment to clarify this point is needed.*

Response: To detect PDZD7 immunostaining, we used a newly generated homemade polyclonal rabbit anti-PDZD7 antibody. It is derived against a mouse PDZD7 fusion protein (amino acid 2-83, accession number NP_001182194.1), and has been validated in transfected cells and mouse organs of Corti. This information is now included in the methods section (Page 21).

We agree with the referee regarding lower number of PDZD7 immunoreactive spots in clarin-2 mutant mice; we have re-written this section to clarify this point. We believe that lack of clarin-2 does not affect the targeting of PDZD7 since all labelled stereocilia display the typical basolateral immunostaining corresponding to the position of the ankle link complex, consistent with previous work (Grati et al. 2012, Zou et al. 2014). The reduced number is due to loss of immunostaining in the regressing short transducing stereocilia, most likely as a result of the disruption of the actin polymerization that follows loss of mechano-electrical transduction activity. We have modified the text to relay these observations (Pages 10 and 14).

Grati M, Shin JB, Weston MD, Green J, Bhat MA, Gillespie PG, Kachar B. Localization of PDZD7 to the stereocilia ankle-link associates this scaffolding protein with the Usher syndrome protein network. *J Neurosci.* 2012 Oct 10;32(41):14288-93. doi: 10.1523/JNEUROSCI.3071-12.2012.

Zou J, Zheng T, Ren C, Askew C, Liu XP, Pan B, Holt JR, Wang Y, Yang J. Deletion of PDZD7 disrupts the Usher syndrome type 2 protein complex in cochlear hair cells and causes hearing loss in mice. *Hum Mol Genet.* 2014 May 1;23(9):2374-90.

3. *There is a confusion in the use of Upper Tip-Link insertion point and Upper Tip-Link Density (UTLD) terms. To the best of my knowledge, the UTL density has never been showed before P10 and consequently can not be used for P6 upper TL insertion point. Therefore the Fig.7c, legend, and methods have to be corrected.*

Response: Following the referee's remark, we now refer to the harmonin-b immunoreactive areas at P6 as the upper attachment point of the tip link (Page 10). During this period, the harmonin-b relocation to this region where cadherin-23 molecules insert into the stereociliary membrane **precedes** the formation of the upper tip link density (UTLD), a very stable structure (easily visible from P10 onwards, see Lefèvre et al. 2008, Grillet et al. 2009, Furness and Hackney, 1985) resulting from the multitude of interactions taking place in the mature stereocilia at the cadherin-23 tip-link component anchor site.

Lefevre G, Michel V, Weil D, Lepelletier L, Bizard E, Wolfrum U, Hardelin JP, Petit C: A core cochlear phenotype in USH1 mouse mutants implicates fibrous links of the hair bundle in its cohesion, orientation and differential growth. *Development* 2008, 135:1427-1437.

Grillet N, Xiong W, Reynolds A, Kazmierczak P, Sato T, Lillo C, Dumont RA, Hintermann E, Sczaniecka A, Schwander M, et al: Harmonin mutations cause mechanotransduction defects in cochlear hair cells. *Neuron* 2009, 62:375-387.

4. *More information about the Clrn2 del629 is needed. What is the consequence of the deletion? A frameshift? Is there a residual protein putatively produced? Looking at the mRNA in the mutant would be essential to define this point.*

Response: We apologise for our oversight in not including a clearer explanation of the *del629* allele in our original submission. We have now included: a description of the allele (Page 5); a schematic of the WT and *del629* alleles (Figure EV1c); and, an RT-PCR and Sanger sequencing of WT and *del629* mutant cochlear RNA (Figure EV1c).

In addition, we provide below ABR, DPOAE and SEM data for the *Cln2^{del629/del629}* mutant mice. These data have not been included in the revised manuscript due to space constraints, and to preserve the flow of the manuscript. However, as you can see, the phenotype of the *del629* mutants closely resembles that of the *clarinet* mutants (i.e. elevated ABR thresholds, reduced DPOAEs, and missing short row stereocilia). Thus, these data strongly suggest that the *del629* mutation, like the *clarinet* mutation, is a loss-of-function allele.

5. In the discussion, the nature and position of the human *CLRN2* variants identified in HL patients are not discussed. This is missing.

Response: We have added additional text to our manuscript to provide details regarding the cluster of SNPs, lying within or close to the *CLRN2* gene, that are significantly associated with an adult hearing difficulty phenotype within the UK Biobank population cohort.

In the Results section (page 6) we have added:

... The second most associated SNP, rs13147559 ($p=1.70E-11$) is in exon 2 of the *CLRN2* gene at coding nucleotide position 337 (c.337, ENST00000511148.2). Presence of the ancestral allele (cytosine, c.337C) encodes for Leucine (p.113Leu), whereas presence of the minor allele (guanine, c.337G) encodes for Valine (p.113Val). As such, this SNP (c.337C>G) represents a missense variant (p.Leu113Val) within the predicted transmembrane domain 2 of the clarin-2 protein (NP_001073296). In silico studies show that the Leucine at position 113 is evolutionarily conserved across species. Furthermore, two prediction tools, PolyPhen-2 and MutationAssessor, suggest that substitution of a Valine at this position might be detrimental to clarin-2 function returning scores of 'possibly damaging' and 'medium', respectively.

In the Discussion section (page 15) we have added:

.... However, the second most associated SNP (rs13147559) is located within the *CLRN2* gene coding sequence (c.337C>G), with presence of the minor allele causing a leucine-to-valine missense variation at codon 113 (p.Leu113Val). This Leucine residue, based on comparison to the 3D modeling prediction of hsCLRN1 (Gyorgy et al. 2019), is located within the second transmembrane domain of hsCLRN2 and is evolutionarily conserved across species. While this variation involves two hydrophobic amino acids that possess similar structures, Valine does have a shorter side-chain. Furthermore, prediction tools suggest that this substitution might be detrimental to protein function returning scores of 'possibly damaging' and 'medium', respectively. Interestingly, similar substitutions located in the highly conserved transmembrane domains of presenilin, encoded by the gene *PSEN1*, have been reported in patients with Alzheimer's disease. These missense variants (p.Leu250Val and p.Leu153Val) have been proposed to interfere with the helix alignment of the transmembrane domain altering protein optimal activity, thus accounting for disease expression (Larner et al. 2013; Furuya et al. 2003). However, additional studies are needed to determine if the hsCLRN2 p.Leu113Val missense variant is causal of, or merely associated with, the adult hearing difficulty trait. Perhaps it may be that 'mild' *CLRN2* hypomorphic mutations, such as p.Leu113Val may represent, is likely to predispose to a progressive, late-onset hearing loss phenotype. Conversely, it is possible that more pathogenic *CLRN2* mutations may elicit a more severe, early-onset hearing loss phenotype. There are examples of this, for instance *TMPRSS3*, encoding Transmembrane Protease Serine 3, has been reported to cause severe-to-profound prelingual hearing loss (DFNB10) as well as progressive hearing impairment with post-lingual onset (DFNB8) due to differential pathogenic mutations (Gao et al. 2017). Of note, a recent work by Gopal S. & colleagues reported a recessively inherited non-syndromic sensorineural hearing loss in a consanguineous Iranian family caused by a *CLRN2* mutation that results in a p.Thr165Lys missense mutation in the encoded protein (Gopal et al. ARO abstract PD 84, p520. 2019). Three affected patients, aged 25, 29 and 44 years old, exhibit moderate-to-profound hearing loss with no indication of balance or vision deficits. Altogether, while mutations in *CLRN1* unambiguously lead to Usher syndrome type IIIA, current findings suggest that *CLRN2* mutation most likely causes non-syndromic hearing loss. However, additional cases need to be identified to clarify the genotype-phenotype relationship between the impaired extent of activity of the mutated clarin-2 protein, the age of onset, the severity, and the extent of the disease phenotype.

Gyorgy B, Meijer EJ, Ivanchenko MV, Tenneson K, Emond F, Hanlon KS, Indzhykulian AA, Volak A, Karavitaki KD, Tamvakologos PI, et al: Gene Transfer with AAV9-PHP.B Rescues Hearing in a Mouse Model of Usher Syndrome 3A and Transduces Hair Cells in a Non-human Primate. *Mol Ther Methods Clin Dev* 2019, 13:1-13.

Larner AJ: Presenilin-1 mutations in Alzheimer's disease: an update on genotype-phenotype relationships. *J Alzheimers Dis* 2013, 37:653-659.

Furuya H, Yasuda M, Terasawa KJ, Tanaka K, Murai H, Kira J, Ohyagi Y: A novel mutation (L250V) in the presenilin 1 gene in a Japanese familial Alzheimer's disease with myoclonus and generalized convulsion. *J Neurol Sci* 2003, 209:75-77.

Gao X, Yuan YY, Wang GJ, Xu JC, Su Y, Lin X, Dai P: Novel Mutations and Mutation Combinations of *TMPRSS3* Cause Various Phenotypes in One Chinese Family with Autosomal Recessive Hearing Impairment. *Biomed Res Int* 2017, 2017:4707315.

Gopal SR, Vona B, Azaiez H, Mazaheri N, Booth KT, Maroofian R, Clancy K, Shariati G, Sedaghat A, Stepanyan R, Smith RJH, Haaf T, Galehdari H, Alagramam KN. (2019) Mutation in the Clarin-2 Gene Cause Hearing Loss in Human and a Zebrafish Model Reveals the Likely Cause of that Hearing Loss. ARO abstract, PD 84, p520.

Minor comments:

1. Concerning the quantification of Harmonin Spot along the tallest row, the sentence in the text lack the reference point (relative to the tip of the taller stereocilium) for the measurements: "The harmonin-b immunoreactive spots were observed on average at 575.4 {plus minus} 23.15 nm (mean {plus minus} s.e.m.; n=35 hair bundles from 5 mice) in *Clrn2clarinet/clarinet* mice, as compared to 850 {plus minus} 28.2 nm (mean {plus minus} s.e.m.; n=31 hair bundles from 5 mice) in *Clrn2clarinet/+* mice (Figure 7f) ($p > 0.0001$, student's t-test)."

Response: The sentence has been replaced by (see page 10 of main text): "The **positioning of the harmonin-b immunoreactive spots, relative to the tip of the taller stereocilium**, were observed on average at 575 ± 23 nm (mean \pm s.e.m.; n=35 hair bundles from 5 mice) in *Clrn2^{clarinet/clarinet}* mice, as

compared to 850 ± 28 nm (mean \pm s.e.m.; n=31 hair bundles from 5 mice) in *Clrn2*^{clarinet/+} mice (Figure 7f) ($p > 0.0001$, student's t-test)."

2. *The gene nomenclature changed the gene name of TMHS to LHFPHL5. The HUGO gene nomenclature Committee still indicated it should be LHFPL5, and therefore this should be corrected.*

Response: This has been corrected.

2nd Editorial Decision

11 July 2019

Thank you for the submission of your revised manuscript to EMBO Molecular Medicine. We have now received the enclosed report from the referees who were asked to re-assess it. As you will see the referees are now overall supportive and I am pleased to inform you that we will be able to accept your manuscript pending the following amendments:

1. please address the ref. #2's comments (summary, discussion and RT-PCR controls) and in writing, answer ref. #3

***** Reviewer's comments *****

Referee #1 (Remarks for Author):

The authors addressed all my questions and comments and revised the manuscript accordingly. Especially, they included novel aspects about human pathogenicity of CLRN2 mutations in the discussion, e.g. by citation of a study from competitors (presented at ARO midwinter meeting). This other study supports the suggestion that CLRN2 mutations are unlikely to cause visual impairments. Furthermore, they experimentally addressed mRNA expression of CLRN2 in the organ of Corti and performed an analysis to quantify the prolateness of the hair bundle tips, as requested by the other reviewers.

Referee #2 (Remarks for Author):

The revised manuscript largely addressed the major concerns, in particular with the data from another study that showed the causative mutations in CLRN2 in human patients. As such the author may want to suggest in the summary that CLRN2 mutations may cause recessive hearing loss.

It will be informative to the community that the authors discuss how their approach of using a large patient cohort could be extended to the study of potential involvement of other deafness genes in humans.

I would suggest that in the Fig.2b to include RT-PCR of an outer hair cell marker in the inner hair cell sample, and an inner hair cell marker in the outer hair cell sample, to rule out any contamination.

Referee #3 (Comments on Novelty/Model System for Author):

This paper describe Clarin-2 as a new deafness gene in mouse and human and place its role in the stereocilia where it is required likely for TL maintenance. Precise localization of the protein (attempted), TL quantification would have improved the model.

Referee #3 (Remarks for Author):

The response letter addressed all my concerns, however one major point indicated as corrected in the manuscript is not: The renaming of UTLD to "upper attachment point of the tip link" for data obtained before P10 this is done in page 10, but not in page 14 neither in the figure 7 c and d and its figure legend.

One new comment is about the abstract: The authors mentioned a "forward genetic screen" but do not indicate in this sentence the species in which it has been done.

Otherwise, congratulation to the team!

2nd Revision - authors' response

23 July 2019

We are very pleased that our manuscript has been accepted for publication in *EMBO Molecular Medicine*. Below is a point-by-point response to the Referees comments regarding the revised manuscript. We believe that we have managed to address all of the concerns raised.

Referee #1:

We thank Referee 1 for their comment - "*The authors addressed all my questions and comments and revised the manuscript accordingly*".

Response: No actions required.

Referee #2:

We thank Referee 2 for their comment - "*The revised manuscript largely addressed the major concerns*".

Comments:

3. *The author may want to suggest in the summary that Cln2 mutations may cause recessive hearing loss.*

Response: We agree with the reviewer, and have included a sentence in the Summary that reads:

'In this study, we establish that Cln2 is a novel deafness gene associated with progressive hearing loss in both mice and humans, and as such severe loss-of-function CLRN2 mutations should be considered in the aetiology of human autosomal recessive hearing loss.'

4. *It will be informative to the community that the authors discuss how their approach of using a large patient cohort could be extended to the study of potential involvement of other deafness genes in humans.*

Response: We agree with the reviewer, and indeed feel that this approach could easily be extended to the interrogation of candidate disease genes more generally, and not just deafness genes. As such, we feel that we should not be too prescriptive, and instead have added a sentence at the end of the Discussion to read:

'Nonetheless, our study demonstrates the utility of interrogating human large cohort study data as a means to help validate candidate genes arising from forward genetic, or whole-genome sequencing, screens.'

3. *I would suggest that in the Fig.2b to include RT-PCR of an outer hair cell marker in the inner hair cell sample, and an inner hair cell marker in the outer hair cell sample, to rule out any contamination.*

Response: Auditory hair cells were isolated under direct visual microscope observation. Only solitary IHCs and OHCs identified based on their typical morphology (cylindrical OHCs and pear-shaped IHCs) were taken into consideration, hair cells with ambiguous morphology were excluded. Furthermore, using the OHC-specific marker, *Oncomodulin (Ocm)*, we obtained no amplification from IHCs, clearly supporting lack of contamination by the outer hair cells. This is now mentioned in the method section, new Figure 2b and related legend.

cylindrical
outer hair cell (OHC)

pear-shaped
inner hair cell (IHC)

Referee #3:

We thank Referee 3 for their comment - "Congratulations to the team!".

Comments:

6. *The renaming of UTLD to "upper attachment point of the tip link" for data obtained before P10 this is done in page 10, but not in page 14 neither in the figure 7 c and d and its figure legend.*

Response: Apologies for this oversight. Changes have now been made to Figure 7, Figure 7 legend and on page 14, so that instead of UTLD we state 'upper attachment point of the tip link (UAPTL)'.

7. *The authors mentioned a "forward genetics screen" but do not indicate in this sentence the species in which it has been done.*

Response: We have added the words "in mice" to this sentence.

Corresponding Author Name: Dr Mike Bowl

Manuscript Number: EMM-2019-10288